# Deep neural networks using a single neuron: folded-in-time architecture using feedback-modulated delay loops

Florian Stelzer[1,2,3], André Röhm[4], Raul Vicente[3], Ingo Fischer [4] & Serhiy Yanchuk [1✉]

Deep neural networks are among the most widely applied machine learning tools showing outstanding performance in a broad range of tasks. We present a method for folding a deep neural network of arbitrary size into a single neuron with multiple time-delayed feedback loops. This single-neuron deep neural network comprises only a single nonlinearity and appropriately adjusted modulations of the feedback signals. The network states emerge in time as a temporal unfolding of the neuron's dynamics. By adjusting the feedback-modulation within the loops, we adapt the network's connection weights. These connection weights are determined via a back-propagation algorithm, where both the delay-induced and local network connections must be taken into account. Our approach can fully represent standard Deep Neural Networks (DNN), encompasses sparse DNNs, and extends the DNN concept toward dynamical systems implementations. The new method, which we call Folded-in-time DNN (Fit-DNN), exhibits promising performance in a set of benchmark tasks.

[1] Institute of Mathematics, Technische Universität Berlin, Berlin, Germany. [2] Department of Mathematics, Humboldt-Universität zu Berlin, Berlin, Germany. [3] Institute of Computer Science, University of Tartu, Tartu, Estonia. [4] Instituto de Física Interdisciplinar y Sistemas Complejos, IFISC (UIB-CSIC), Campus Universitat de les Illes Baleares, Palma de Mallorca, Spain. ✉email: yanchuk@math.tu-berlin.de

Fueled by deep neural networks (DNN), machine learning systems are achieving outstanding results in large-scale problems. The data-driven representations learned by DNNs empower state-of-the-art solutions to a range of tasks in computer vision, reinforcement learning, robotics, healthcare, and natural language processing[1–9]. Their success has also motivated the implementation of DNNs using alternative hardware platforms, such as photonic or electronic concepts, see, e.g., refs. [10–12] and references therein. However, so far, these alternative hardware implementations require major technological efforts to realize partial functionalities, and, depending on the hardware platform, the corresponding size of the DNN remains rather limited[12].

Here, we introduce a folding-in-time approach to emulate a full DNN using only a single artificial neuron with feedback-modulated delay loops. Temporal modulation of the signals within the individual delay loops allows realizing adjustable connection weights among the hidden layers. This approach can reduce the required hardware drastically and offers a new perspective on how to construct trainable complex systems: The large network of many interacting elements is replaced by a single element, representing different elements in time by interacting with its own delayed states. We are able to show that our folding-in-time approach is fully equivalent to a feed-forward deep neural network under certain constraints—and that it, in addition, encompasses dynamical systems specific architectures. We name our approach Folded-in-time Deep Neural Network or short Fit-DNN.

Our approach follows an interdisciplinary mindset that draws its inspiration from the intersection of AI systems, brain-inspired hardware, dynamical systems, and analog computing. Choosing such a different perspective on DNNs leads to a better understanding of their properties, requirements, and capabilities. In particular, we discuss the nature of our Fit-DNN from a dynamical systems' perspective. We derive a back-propagation approach applicable to gradient descent training of Fit-DNNs based on continuous dynamical systems and demonstrate that it provides good performance results in a number of tasks. Our approach will open up new strategies to implement DNNs in alternative hardware.

For the related machine learning method called "reservoir computing" based on fixed recurrent neural networks, folding-in-time concepts have already been successfully developed[13]. Delay-based reservoir computing typically uses a single delay loop configuration and time-multiplexing of the input data to emulate a ring topology. The introduction of this concept led to a better understanding of reservoir computing, its minimal requirements, and suitable parameter conditions. Moreover, it facilitated their implementation on various hardware platforms[13–19]. In fact, the delay-based reservoir computing concept inspired successful implementations in terms of hardware efficiency[13], processing speed[16,20,21], task performance[22,23], and last, but not least, energy consumption[16,22].

Our concept of folded-in-time deep neural networks also benefits from time-multiplexing, but uses it in a more intricate manner going conceptually beyond by allowing for the implementation of multi-layer feed-forward neural networks with adaptable hidden layer connections and, in particular, the applicability of the gradient descent method for their training. We present the Fit-DNN concept and show its versatility and applicability by solving benchmark tasks.

## Results

### A network folded into a single neuron.
The traditional Deep Neural Networks consist of multiple layers of neurons coupled in a feed-forward architecture. Implementing their functionality with only a single neuron requires preserving the logical order of the layers while finding a way to sequentialize the operation within the layer. This can only be achieved by temporally spacing out processes that previously acted simultaneously. A single neuron receiving the correct inputs at the correct times sequentially emulates each neuron in every layer. The connections that previously linked neighboring layers now instead have to connect the single neuron at different times, and thus interlayer links turn into delay-connections. The weight of these connections has to be adjustable, and therefore a temporal modulation of these connections is required.

The architecture derived this way is depicted in Fig. 1 and called Folded-in-time DNN. The core of the Fit-DNN consists of a single neuron with multiple delayed and modulated feedbacks. The type or exact nature of the single neuron is not essential. To facilitate the presentation of the main ideas, we assume that the system state evolves in continuous time according to a differential equation of the general form:

$$\dot{x}(t) = -\alpha x(t) + f(a(t)), \quad \text{where} \tag{1}$$

$$a(t) = J(t) + b(t) + \sum_{d=1}^{D} \mathcal{M}_d(t) x(t - \tau_d). \tag{2}$$

Here $x(t)$ denotes the state of the neuron; $f$ is a nonlinear function with the argument $a(t)$ combining the data signal $J(t)$, time-varying bias $b(t)$, and the time-delayed feedback signals $x(t - \tau_d)$ modulated by the functions $\mathcal{M}_d(t)$, see Fig. 1. We explicitly consider multiple loops of different delay lengths $\tau_d$. Due to the feedback loops, the system becomes a so-called delay dynamical system, which leads to profound implications for the complexity of its dynamics[24–32]. Systems of the form (1) are typical for machine learning applications with delay models[13,14,20,33].

Intuitively, the feedback loops in Fig. 1 lead to a reintroduction of information that has already passed through the nonlinearity $f$. This allows chaining the nonlinearity $f$ many times. While a classical DNN composes its trainable representations by using neurons layer-by-layer, the Fit-DNN achieves the same by reintroducing a feedback signal to the same neuron repeatedly.

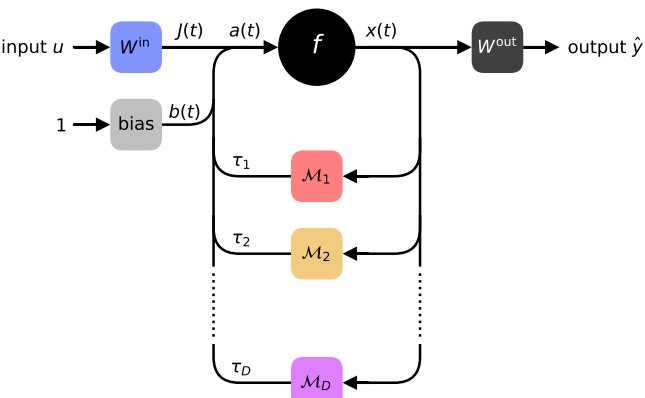

**Fig. 1 Scheme of the Fit-DNN setup.** A nonlinear element (neuron) with a nonlinear function $f$ is depicted by a black circle. The state of the neuron at time $t$ is $x(t)$. The signal $a(t)$ is the sum of the data $J(t)$, bias $b(t)$, and feedback signals. Adjustable elements are indicated by square boxes. The data signal is generated from an input vector **u** using a matrix $W^{\text{in}}$ containing input weights (blue box). The bias signal is generated using bias coefficients (light gray box). Each feedback loop implements a delay $\tau_d$ and a temporal modulation $\mathcal{M}_d(t)$ (color boxes) to generate the feedback signals. Finally, the output is obtained from the signal $x(t)$ using a matrix $W^{\text{out}}$ of output weights (dark gray box).

In each pass, the time-varying bias $b(t)$ and the modulations $\mathcal{M}_d(t)$ on the delay-lines ensure that the time evolution of the system processes information in the desired way. To obtain the data signal $J(t)$ and output $\hat{y}$ we need an appropriate pre- or postprocessing, respectively.

**Equivalence to multi-layer neural networks.** To further illustrate how the Fit-DNN is functionally equivalent to a multi-layer neural network, we present Fig. 2 showing the main conceptual steps for transforming the dynamics of a single neuron with multiple delay loops into a DNN. A sketch of the time-evolution of $x(t)$ is presented in Fig. 2a. This evolution is divided into time-intervals of length $T$, each emulating a hidden layer. In each of the intervals, we choose $N$ points. We use a grid of equidistant timings with small temporal separation $\theta$. For hidden layers with $N$ nodes, it follows that $\theta = T/N$. At each of these temporal grid points $t_n = n\theta$, we treat the system state $x(t_n)$ as an independent variable. Each temporal grid point $t_n$ will represent a node, and $x(t_n)$ its state. We furthermore assume that the data signal $J(t)$, bias $b(t)$, and modulation signals $\mathcal{M}_d(t)$ are step functions with step-lengths $\theta$; we refer to the "Methods" section for their precise definitions.

By considering the dynamical evolution of the time-continuous system $x(t)$ only at these discrete temporal grid points $t_n$ (black dots in Fig. 2a), one can prove that the Fit-DNN emulates a classical DNN. To show it formally, we define network nodes $x_n^\ell$

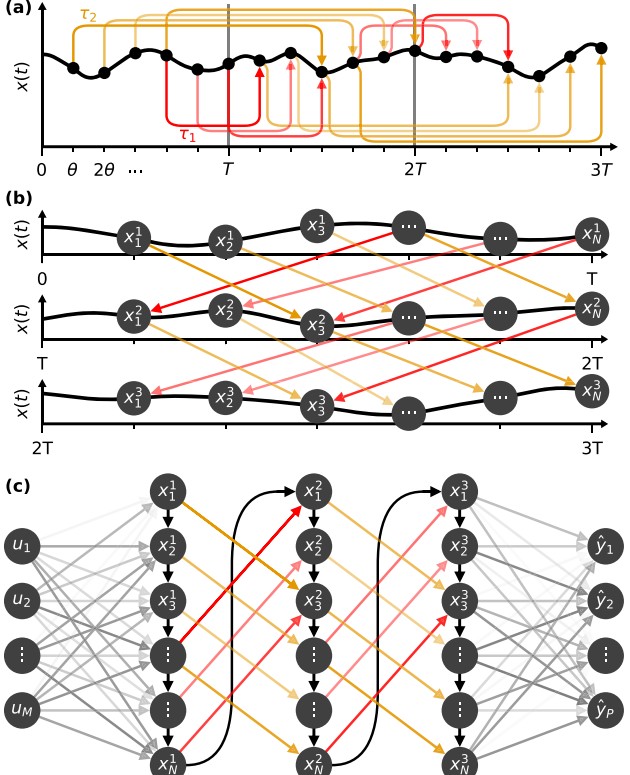

**Fig. 2 Equivalence of the Fit-DNN using a single neuron with modulated delayed feedbacks to a classical DNN. a** The neuron state is considered at discrete time points $x_n^\ell := x((\ell - 1)T + n\theta)$. The intervals $((\ell - 1)T, \ell T]$ correspond to layers. Due to delayed feedbacks, non-local connections emerge (color lines). **b** Shows a stacked version of the plot in (**a**) with the same active connections. **c** Shows the resulting network: it is a rotated version of (**b**), with additional input and output layers. Black lines indicate connections implied by the temporal ordering of the emulation.

of the equivalent DNN as

$$x_n^\ell := x((\ell - 1)T + n\theta), \qquad (3)$$

with $n = 1, \ldots, N$ determining the node's position within the layer, and $\ell = 1, \ldots, L$ determining the layer. Analogously, we define the activations $a_n^\ell$ of the corresponding nodes. Furthermore, we add an additional node $x_{N+1}^\ell := 1$ to take into account the bias. Thus, the points from the original time-intervals $T$ are now described by the vector $\mathbf{x}^\ell = (x_1^\ell, \ldots, x_N^\ell)$. Fig. 2b shows the original time-trace cut into intervals of length $T$ and nodes labeled according to their network position. The representation in Fig. 2c is a rotation of Fig. 2b with the addition of an input and an output layer.

The connections are determined by the dynamical dependencies between the nodes $x_n^\ell$. These dependencies can be explicitly calculated either for small or large distance $\theta$. In the case of a large node separation $\theta$, the relations between the network nodes $x_n^\ell$ is of the familiar DNN shape:

$$x_n^\ell = \alpha^{-1} f(a_n^\ell), \qquad (4)$$

$$\mathbf{a}^\ell := W^\ell \mathbf{x}^{\ell-1}. \qquad (5)$$

System (4) is derived in detail in the Supplementary Information. The matrix $W^\ell$ describes the connections from layer $\ell - 1$ to $\ell$ and corresponds to the modulated delay-lines in the original single-neuron system. Each of the time-delayed feedback loops leads to a dependence of the state $x(t)$ on $x(t - \tau_d)$, see colored arrows in Fig. 2a. By way of construction, the length of each delay-loop is fixed. Since the order of the nodes (3) is tied to the temporal position, a fixed delay-line cannot connect arbitrary nodes. Rather, each delay-line is equivalent to one diagonal of the coupling matrix $W^\ell$. Depending on the number of delay loops $D$, the network possesses a different connectivity level between the layers. A fully connected Fit-DNN requires $2N - 1$ modulated delay loops, i.e., our connectivity requirement scales linearly in the system size $N$ and is entirely independent of $L$, promising a favorable scaling for hardware implementations.

The time-dependent modulation signals $\mathcal{M}_d(t)$ allow us to set the feedback strengths to zero at certain times. For this work, we limit ourselves to delayed feedback connections, which only link nodes from the neighboring layers, but in principle this limitation could be lifted if more exotic networks were desired. For a visual representation of the connections implied by two sample delay loops, see Fig. 2b and c. The mismatch between the delay $\tau_d$ and $T$ determines, which nodes are connected by that particular delay-loop: For $\tau_d < T$ ($\tau_d > T$), the delayed feedback connects a node $x_n^\ell$ with another node $x_i^{\ell+1}$ in a subsequent layer with $n > i$ ($n < i$), shown with red (yellow) arrows in Fig. 2.

To complete the DNN picture, the activations for the first layer will be rewritten as $\mathbf{a}^1 := g(\mathbf{a}^{in}) := g(W^{in}\mathbf{u})$, where $W^{in}$ is used in the preprocessing of $J(t)$. A final output matrix $W^{out}$ is used to derive the activations of the output layer $\mathbf{a}^{out} := W^{out}\mathbf{x}^L$. We refer to the "Methods" section for a precise mathematical description.

**Dynamical systems perspective: small node separation.** For small node separation $\theta$, the Fit-DNN approach goes beyond the standard DNN. Inspired by the method used in refs. [13,34,35], we apply the variation of constants formula to solve the linear part of (1) and the Euler discretization for the nonlinear part and obtain the following relations between the nodes up to the first-order terms in $\theta$:

$$x_n^\ell = e^{-\alpha\theta} x_{n-1}^\ell + \alpha^{-1}(1 - e^{-\alpha\theta}) f(a_n^\ell), \quad n = 2, \ldots, N, \qquad (6)$$

for the layers $\ell = 1, \ldots, L$, and nodes $n = 2, \ldots, N$. Note, how the

first term $e^{-\alpha\theta}x_{n-1}^{\ell}$ couples each node to the preceding one within the same layer. Furthermore, the first node of each layer $\ell$ is connected to the last node of the preceding layer:

$$x_1^{\ell} = e^{-\alpha\theta}x_N^{\ell-1} + \alpha^{-1}(1 - e^{-\alpha\theta})f(a_1^{\ell}), \qquad (7)$$

where $x_N^0 := x_0 = x(0)$ is the initial state of system (1). Such a dependence reflects the fact that the network was created from a single neuron with time-continuous dynamics. With a small node separation $\theta$, each node state residually depends on the preceding one and is not fully independent. These additional 'inertial' connections are represented by the black arrows in the network representation in Fig. 2c and are present in the case of small $\theta$.

This second case of small $\theta$ may seem like a spurious, superfluous regime that unnecessarily complicates the picture. However, in practice, a small $\theta$ directly implies a fast operation—as the time the single neuron needs to emulate a layer is directly given by $N\theta$. We, therefore, expect this regime to be of interest for future hardware implementations. Additionally, while we recover a fully connected DNN using $D = 2N - 1$ delay loops, our simulations show that this is not a strict requirement. Adequate performance can already be obtained with a much smaller number of delay loops. In that case, the Fit-DNN is implementing a particular type of sparse DNNs.

**Back-propagation for Fit-DNN.** The Fit-DNN (4) for large $\theta$ is the classical multilayer perceptron; hence, the weight gradients can be computed using the classical back-propagation algorithm[3,36,37]. If less than the full number of delay-loops is used, the resulting DNN will be sparse. Training sparse DNN is a current topic of research[38,39]. However, the sparsity does not affect the gradient computation for the weight adaptation.

For a small temporal node separation $\theta$, the Fit-DNN approach differs from the classical multilayer perceptron because it contains additional linear intra-layer connections and additional linear connections from the last node of one hidden layer to the first node of the next hidden layer, see Fig. 2c, black arrows. Nonetheless, the network can be trained by adjusting the input weights $W^{\text{in}}$, the output weights $W^{\text{out}}$, and the non-zero elements of the potentially sparse weight matrices $W^{\ell}$ using gradient descent. For this, we employ a back-propagation algorithm, described in section "Application to machine learning and a back-propagation algorithm", which takes these additional connections into consideration.

**Benchmark tasks.** Since under certain conditions, the Fit-DNN fully recovers a standard DNN (without convolutional layers), the resulting performance will be identical. This is obvious, when considering system (4), since the dynamics are perfectly described by a standard multilayer perceptron. However, the Fit-DNN approach also encompasses the aforementioned cases of short temporal node distance $\theta$ and the possibility of using less delay-loops, which translates to a sparse DNN. We report here that the

system retains its computational power even in these regimes, i.e., a Fit-DNN can in principle be constructed with few and short delay-loops.

To demonstrate the computational capabilities of the Fit-DNN over these regimes, we considered five image classification tasks: MNIST[40], Fashion-MNIST[41], CIFAR-10, CIFAR-100 considering the coarse class labels[42], and the cropped version of SVHN[43]. As a demonstration for a very sparse network, we applied the Fit-DNN to an image denoising task: we added Gaussian noise of intensity $\sigma_{\text{task}} = 1$ to the images of the Fashion-MNIST dataset, which we considered as vectors with values between 0 (white) and 1 (black). Then we clipped the resulting vector entries at the clipping thresholds 0 and 1 in order to obtain noisy grayscale images. The denoising task is to reconstruct the original images from their noisy versions. Fig. 3 shows examples of the original Fashion-MNIST images, their noisy versions, and reconstructed images.

For the tests, we solved the delay system (1) numerically and trained the weights by gradient descent using the back-propagation algorithm described in the section. "Application to machine learning and a back-propagation algorithm". Unless noted otherwise, we operated in the small $\theta$ regime, and in general did not use a fully connected network. By nature of the architecture, the choice of delays $\tau_d$ is not trivial. We always chose the delays as a multiple of $\theta$, i.e. $\tau_d = n_d\theta$, $d = 1, \ldots, D$. The integer $n_d$ can range from 1 to $2N - 1$ and indicates which diagonal of the weight matrix $W^{\ell}$ is accessed. After some initial tests, we settled on drawing the numbers $n_d$ from a uniform distribution on the set $\{1, \ldots, 2N - 1\}$ without replacement.

If not stated otherwise, we used the activation function $f(a) = \sin(a)$, but the Fit-DNN is in principle agnostic to the type of nonlinearity $f$ that is used. The standard parameters for our numerical tests are listed in Table 1. For further details, we refer to the "Methods" "Data augmentation, input processing and initialization" section.

In Table 2, we show the Fit-DNN performance for different numbers of the nodes $N = 50, 100, 200$, and $400$ per hidden layer on the aforementioned tasks. We immediately achieve high success rates on the relatively simple MNIST and Fashion-MNIST tasks. The more challenging CIFAR-10, coarse CIFAR-100, and cropped SVHN tasks obtain lower yet still significant success rates. The confusion matrices (see Supplementary Information) also show that the system tends to confuse similar categories (e.g., "automobile" and "truck"). While these results clearly do not rival record state-of-the art performances, they were achieved on a novel and radically different architecture. In particular, the Fit-DNN here only used about half of the available diagonals of the weight matrix and operated in the small $\theta$ regime. For the tasks tested, increasing $N$ clearly leads to increased performance. This also serves as a sanity check and proves the scalability of the concept. In particular, note that if implemented in some form of dedicated hardware, increasing the number of nodes per layer $N$ does not increase the number of components needed, solely the time required to run the system.

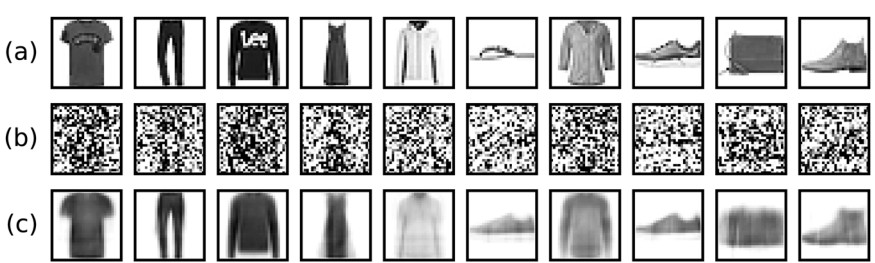

**Fig. 3 Example images for the denoising task.** Row **a** contains original images from the Fashion-MNIST dataset. Row **b** shows the same images with additional Gaussian noise. These noisy images serve as input data for the trained system. Row **c** shows the obtained reconstructions of the original images.

**Table 1 Default parameters.**

|  | (a) | (b) | (c) | (d) |
|---|---|---|---|---|
| input nodes $M$ | 784 | 3072 | 3072 | 784 |
| output nodes $P$ | 10 | 10 | 20 | 784 |
| nodes per hidden layer $N$ | 100 | 100 | 100 | 100 |
| number of hidden layers $L$ | 2 | 3 | 3 | 2 |
| number of delays $D$ | 100 | 100 | 100 | 5 |
| node separation $\theta$ | 0.5 | 0.5 | 0.5 | 0.5 |
| system time scale $\alpha$ | 1 | 1 | 1 | 1 |
| initial training rate $\eta_0$ | 0.01 | 0.0001 | 0.0001 | 0.001 |
| training rate scaling factor $\eta_1$ | 10000 | 1000 | 1000 | 500 |
| intensity of training noise $\sigma$ | 0.1 | 0.01 | 0.01 | – |

Standard parameters for (a) the MNIST and Fashion-MNIST tasks, (b) the CIFAR-10 and cropped SVHN tasks, (c) the CIFAR-100 tasks with coarse class labels, and (d) the image denoising task.

**Table 2 Fit-DNN performance for classification and denoising tasks depending on the number of nodes per hidden layer $N$.**

| $N$ | 50 | 100 | 200 | 400 | |
|---|---|---|---|---|---|
| MNIST | 97.31 | 98.49 | 98.91 | 98.97 | [%] |
| Fashion-MNIST | 86.61 | 87.82 | 88.59 | 89.18 | [%] |
| CIFAR-10 | 48.29 | 51.42 | 53.94 | 54.99 | [%] |
| Coarse CIFAR-100 | 29.39 | 32.73 | 34.51 | 35.41 | [%] |
| Cropped SVHN | 73.45 | 78.93 | 80.85 | 81.38 | [%] |
| Denoising | 0.0277 | 0.0254 | 0.0241 | 0.0236 | [MSE] |

Shown are accuracies [in %] and mean squared error for the denoising task for different $N$. Increasing $N$ improves the results for all tasks. For the classification tasks with $N = 50$, the number of delays is $D = 99$, for the other cases the standard value $D = 100$ is used. For the denoising task, $D = 5$ is used for all cases.

Also note, that the denoising task was solved using only 5 delay-loops. For a network of 400 nodes, this results in an extremely sparse weight matrix $W^\ell$. Nonetheless, the system performs well.

Figure 4 shows the performance of the Fit-DNN for the classification tasks and the correctness of the computed gradients for different node separations $\theta$. Since this is one of the key parameters that controls the Fit-DNN, understanding its influences is of vital interest. We also use this opportunity to illustrate the importance of considering the linear local connections when performing back-propagation to compute the weight gradients. We applied gradient checking, i.e., the comparison to a numerically computed practically exact gradient, to determine the correctness of the obtained gradient estimates. We also trained the map limit network (4) for comparison, corresponding to a (sparse) multilayer perceptron. In this way, we can also see how the additional intra-layer connections influence the performance for small $\theta$.

The obtained results of Fig. 4 show that back-propagation provides good estimates of the gradient over the entire range of $\theta$. They also highlight the strong influence of the local connections. More specifically, taking into account the local connections, the back-propagation algorithm yields correct gradients for large node separations $\theta \geq 4$ and for small node separations $\theta \leq 0.125$ (blue points in Fig. 4). For intermediate node separations, we obtain a rather rough approximation of the gradient, but the cosine similarity between the actual gradient and its approximation is still at least 0.8, i.e., the approximation is good enough to train effectively. In contrast, if local connections are neglected, back-propagation works only for a large node separation $\theta \geq 4$, where the system approaches the map limit (red points in Fig. 4).

Consequently, we obtain competitive accuracies for the MNIST and the Fashion-MNIST tasks even for small $\theta$ if we use back-propagation with properly included local connections. When we apply the Fit-DNN to the more challenging CIFAR-10, coarse CIFAR-100, and cropped SVHN tasks, small node separations affect the accuracies negatively. However, we still obtain reasonable results for moderate node separations.

Further numerical results regarding the number of hidden layers $L$, the number of delays $D$, and the role of the activation function $f$ are presented in detail in the Supplementary Information. We find that the optimal choice of $L$ depends on the node separation $\theta$. Our findings suggest that for small $\theta$, one should choose a smaller number of hidden layers than for the map limit case $\theta \to \infty$. The effect of the number of delays $D$ depends on the task. We found that a small number of delays is sufficient for the denoising task: the mean squared error remains constant when varying $D$ between 5 and 40. For the CIFAR-10 task, a larger number of delays is necessary to obtain optimal results. If we use the standard parameters from Table 1, we obtain the highest CIFAR-10 accuracy for $D = 125$ or larger. This could likely be explained by the different requirements of these tasks: While the main challenge for denoising is to filter out unwanted points, the CIFAR-10 task requires attention to detail. Thus, a higher number of delay-loops potentially helps the system to learn a more precise representation of the target classes. By comparing the Fit-DNN performance for different activation functions, we also confirmed that the system performs similarly well for the sine $f(a) = \sin(a)$, the hyperbolic tangent $f(a) = \tanh(a)$, and the ReLU function $f(a) = \max\{0, a\}$.

## Discussion

We have designed a method for complete folding-in-time of a multilayer feed-forward DNN. This Fit-DNN approach requires only a single neuron with feedback-modulated delay loops. Via a temporal sequentialization of the nonlinear operations, an arbitrarily deep or wide DNN can be realized. We also naturally arrive at such modifications as sparse DNNs or DNNs with additional inertial connections. We have demonstrated that gradient descent training of the coupling weights is not significantly interfered by these additional local connections.

Extending machine-learning architectures to be compatible with a dynamical delay-system perspective can help fertilize both fundamental research and applications. For example, the idea of time-multiplexing a recurrent network into a single element was introduced in ref. [13] and had a profound effect on understanding and boosting the reservoir computing concept. In contrast to the time-multiplexing of a fixed recurrent network for reservoir computing, here we use the extended folding-in-time technique to realise feed-forward DNNs, thus implementing layers with

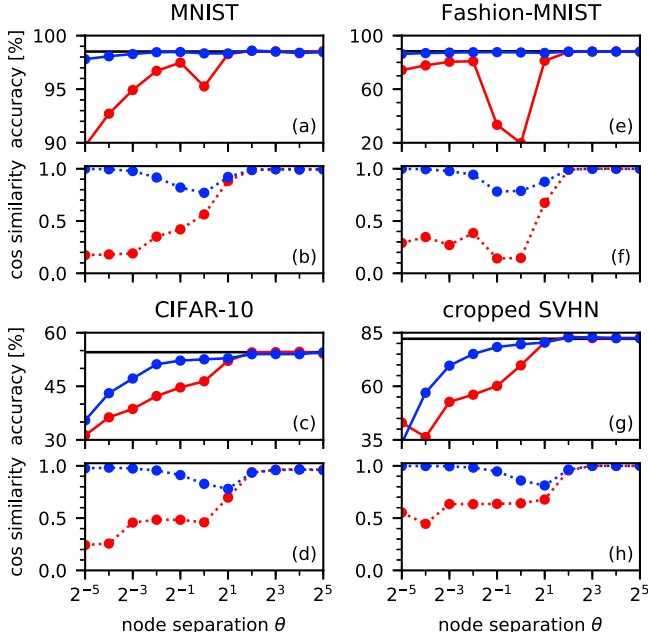

**Fig. 4 Fit-DNN performance for classification and denoising tasks; dependence on the node separation θ.** Shown are accuracies of the classification tasks by employing the back-propagation algorithm taking the local coupling into consideration (blue points), and neglecting them (red points); (**a**, **c**, **e**, and **g**). The accuracy obtained in the map limit case θ → ∞ is shown by the horizontal black line (this corresponds to the classical sparse multilayer perceptron). Lower panels **b**, **d**, **f**, and **h** show the cosine similarities between the numerically computed approximation of the exact gradient and the gradient obtained by back-propagation with (blue points) or without (red) local connections.

adaptive connection weights. Compared to delay-based reservoir computing, our concept focuses on the different and extended range of possible applications of DNNs.

From a general perspective, our approach provides an alternative view on neural networks: the entire topological complexity of the feed-forward multilayer neural networks can be folded into the temporal domain by the delay-loop architecture. This exploits the prominent advantage of time-delay systems that "space" and "time" can intermingle, and delay systems are known to have rich spatio-temporal properties[32,44–46]. This work significantly extends this spatio-temporal equivalence and its application while allowing the evaluation of neural networks with the tools of delay systems analysis[26,30,47,48]. In particular, we show how the transition from the time-continuous view of the physical system, i.e., the delay-differential equation, to the time-discrete feed-forward DNN can be made.

Our concept also differs clearly from the construction of neural networks from ordinary differential equations[49–51]. Its main advantage is that delay systems inherently possess an infinite-dimensional phase space. As a result, just one neuron with feedback is sufficient to fold the entire complexity of the network.

It has been shown that dynamic sparsity[38,39] can outperform dense networks and, fundamentally, Fit-DNNs are intrinsically compatible with certain kinds of sparsity. However, in our approach, removing or adding a delay loop would change an entire diagonal in the hidden weight matrices. Therefore, sparsity training algorithms such as those discussed in refs.[38,39] and related works are not directly applicable to the Fit-DNN. Our preliminary tests have shown that removing the weights of a diagonal at the same time disturbs the previous training too much, so the method fails. Nevertheless, we expect that it is

possible to find a suitable method to optimize the choice of delays. Therefore, further investigation of specific sparsity training methods for the Fit-DNN would be very welcome. One candidate for such a method could be pruning by slowly fading diagonals that contain weaker connections on average.

Even with a fixed sparse connectivity, we can perform image classification using only a single dynamical neuron. This case, in particular, highlights one of the most exciting aspects of the Fit-DNN architecture: many hardware implementations of DNNs or related systems have suffered from the large amount of elements that need to be implemented: the active neurons as well as the connections with adjustable weights. The Fit-DNN overcomes both of these limitations; no matter how many neurons are functionally desired, physically we only require a single one. Even though we advocate for sparse connectivity in this paper, a fully connected DNN would only require a linear scaling of the number of delay loops with the number of nodes per layer N. This represents a major advantage as compared to directly implemented networks, where the number of connections grows quadratically. Thus, where it is acceptable to use sparse networks, increasing the number of layers L or the number of nodes per layer N for the Fit-DNN only requires more time, but not more hardware elements.

Another major aspect of the Fit-DNN construction is the importance of the temporal node separation θ. For large node separation θ, the Fit-DNN mimics conventional multilayer perceptrons. Therefore, the performance in terms of accuracy is equivalent in this case. In contrast, choosing a smaller θ benefits the overall computation time, but decreases the achievable accuracy. This decrease strongly depends on the considered tasks (see Fig. 4).

In addition to providing a dynamical systems perspective on DNNs, Fit-DNNs can also serve as blueprints for specialized DNN hardware. The Fit-DNN approach is agnostic concerning the type of nonlinearity, enabling flexibility of implementations. A suitable candidate could be a photonic neuromorphic implementation[13–16,20,52,53], where a fast artificial neuron can be realized with the Gigahertz timescale range. Photonic systems have already been used to construct delay-based reservoir computers. In retrospect, it is quite clear how instrumental the reduced hardware requirement of a delay-based approach was in stimulating the current ecosystem of reservoir computing implementations. For example, the delay-based reservoir computing has been successfully implemented using electronic systems, magnetic spin systems, MEMS, acoustic, and other platforms. We hope that for the much larger community around DNNs, a similarly stimulating effect can be achieved with the Fit-DNN approach we presented here, since it also drastically reduces the cost and complexity for hardware-based DNNs.

Certainly, realizations on different hardware platforms face different challenges. In the following, we exemplify the requirements for a photonic (optoelectronic) scheme. Such an implementation requires only one light source, a few fiber couplers, and optical fibers of different lengths. The modulations of the delay loops can be implemented using Mach-Zehnder intensity modulators. Finally, only two fast photodetectors (one for all delay loops and one for the output) would be required, as well as an optical amplifier or an electrical amplifier which could be used to compensate for roundtrip losses. Those are all standard telecommunication components. The conversion from optical to electrical signals can be done extremely fast, faster than the clock rate of today's fast electronic processors, and only two photodetectors are needed, regardless of the number of virtual nodes and number of delay loops.

The Fit-DNN setup allows to balance between computational speed and the number of required hardware components. Since

only one nonlinear node and one fast read-out element are absolutely necessary in our approach, ultrafast components could be used that would be unrealistic or too expensive for full DNN implementations. At the same time, since the single nonlinear element performs all nonlinear operations sequentially with node separation $\theta$, parallelization cannot be applied in this approach. The overall processing time scales linearly with the total number of nodes $LN$ and with the node separation $\theta$. Possible ways to address this property that could represent a limitation in certain applications include the use of a small node separation $\theta$[13] or multiple parallel copies of Fit-DNNs. In this way, a tradeoff between the number of required hardware components and the amount of parallel processing is possible. At the same time, the use of a single nonlinear node comes with the advantage of almost perfect homogeneity of all folded nodes, since they are realised by the same element.

We would also like to point out that the potential use of very fast hardware components is accompanied by a possibility of fast inference. However, a fast hardware implementation of the Fit-DNN will not accelerate the training process, because a traditional computer is still required, at least for the back-propagation of errors. If the forward propagation part of the training process is also performed on a traditional computer, the delay equation must be solved numerically for each training step, leading to a significant increase in training time. Therefore, the presented method is most suitable when fast inference and/or high hardware efficiency are prioritized. We would like to point out that the integration of the training process into the hardware-part could be addressed in future extensions of our concept.

We have presented a minimal and concise model, but already a multitude of potential extensions are apparent for future studies. For instance, one can implement different layer sizes, multiple nonlinear elements, and combine different structures such as recurrent neural networks with trainable hidden layers.

Incorporating additional neurons (spatial nodes) might even enable finding the optimal trade-off between spatial and temporal nodes, depending on the chosen platform and task. Also, we envision building a hierarchical neural network consisting of interacting neurons, each of them folding a separate Fit-DNN in the temporal domain. Altogether, starting with the design used in this work, we might unlock a plethora of neural network architectures.

Finally, our approach encourages further cross-fertilization among different communities. While the spatio-temporal equivalence and the peculiar properties of delay-systems may be known in the dynamical systems community, so far, no application to DNNs had been considered. Conversely, the Machine Learning core idea is remarkably powerful, but usually not formulated to be compatible with continuous-time delay-dynamical systems. The Fit-DNN approach unifies these perspectives—and in doing so, provides a concept that is promising for those seeking a different angle to obtain a better understanding or to implement the functionality of DNNs in dedicated hardware.

## Methods

### The delay system and the signal *a(t)*.
The delay system (1) is driven by a signal $a(t)$ which is defined by Eq. (2) as a sum of a data signal $J(t)$, modulated delayed feedbacks $\mathcal{M}_d(t)x(t - \tau_d)$, and a bias $b(t)$. In the following, we describe the components in detail.

(i) *The input signal.* Given an input vector $(u_1, \dots, u_M)^{\mathrm{T}} \in \mathbb{R}^M$, a matrix $W^{\mathrm{in}} \in \mathbb{R}^{N \times (M+1)}$ of input weights $w^{\mathrm{in}}_{nm}$ and an input scaling function $g$, we define

$$J(t) := g\left(w^{\mathrm{in}}_{n,M+1} + \sum_{m=1}^{M} w^{\mathrm{in}}_{nm} u_m\right), \qquad (8)$$

for $(n-1)\theta < t \le n\theta$ and $n = 1, \dots, N$. This rule defines the input signal $J(t)$ on the time interval $(0, T]$, whereas $J(t) = 0$ for the other values of $t$. Such a

restriction ensures that the input layer connects only to the first hidden layer of the Fit-DNN. Moreover, $J(t)$ is a step function with the step lengths $\theta$.

(ii) *The feedback signals.* System (1) contains $D$ delayed feedback terms $\mathcal{M}_d(t)x(t - \tau_d)$ with the delay times $\tau_1 < \dots < \tau_D$, which are integer multiples of the stepsize $\tau_d = n_d\theta$, $n_d \in \{1, \dots, 2N-1\}$. The modulation functions $\mathcal{M}_d$ are defined interval-wise on the layer intervals $((\ell-1)T, \ell T]$. In particular, $\mathcal{M}_d(t) := 0$ for $t \le T$. For $(\ell-1)T + (n-1)\theta < t \le (\ell-1)T + n\theta$ with $\ell = 2, \dots, L$ and $n = 1, \dots, N$, we set

$$\mathcal{M}_d(t) := v^{\ell}_{d,n}. \qquad (9)$$

Thus, the modulation functions $\mathcal{M}_d(t)$ are step functions with step length $\theta$. The numbers $v^{\ell}_{d,n}$ play the role of the connection weights from layer $\ell - 1$ to layer $\ell$. More precisely, $v^{\ell}_{d,n}$ is the weight of the connection from the $(n + N - n_d)$-th node of layer $\ell - 1$ to the $n$-th node of layer $\ell$. Section "Network representation for small node separation $\theta$" below explains how the modulation functions translate to the hidden weight matrices $W^{\ell}$. In order to ensure that the delay terms connect only consecutive layers, we set $v^{\ell}_{d,n} = 0$ whenever $n_d < n$ or $n_d > n + N - 1$ holds.

(iii) *The bias signal.* Finally, the bias signal $b(t)$ is defined as the step function

$$b(t) := b^{\ell}_n, \qquad \text{for } (\ell-1)T + (n-1)\theta < t \le (\ell-1)T + n\theta, \qquad (10)$$

where $n = 1, \dots, N$ and $\ell = 2, \dots, L$. For $0 \le t \le T$, we set $b(t) := 0$ because the bias weights for the first hidden layer are already included in $W^{\mathrm{in}}$, and thus in $J(t)$.

### Network representation for small node separation *θ*.
In this section, we provide details to the network representation of the Fit-DNN which was outlined in "Results" section. The delay system (1) is considered on the time interval $[0, LT]$. As we have shown in "Results" section, it can be considered as multi-layer neural network with $L$ hidden layers, represented by the solution on sub-intervals of length $T$. Each of the hidden layers consists of $N$ nodes. Moreover, the network possesses an input layer with $M$ nodes and an output layer with $P$ nodes. The input and hidden layers are derived from the system (1) by a discretization of the delay system with step length $\theta$. The output layer is obtained by a suitable readout function on the last hidden layer.

We first construct matrices $W^{\ell} = (w^{\ell}_{nj}) \in \mathbb{R}^{N \times (N+1)}$, $\ell = 2, \dots, L$, containing the connection weights from layer $\ell - 1$ to layer $\ell$. These matrices are set up as follows: Let $n'_d := n_d - N$, then $w^{\ell}_{n,n-n'_d} := v^{\ell}_{d,n}$ define the elements of the matrices $W^{\ell}$. All other matrix entries (except the last column) are defined to be zero. The last column is filled with the bias weights $b^{\ell}_1, \dots, b^{\ell}_N$. More specifically,

$$w^{\ell}_{nj} := \delta_{N+1,j} b^{\ell}_n + \sum_{d=1}^{D} \delta_{n-n'_d,j} v^{\ell}_{d,n}, \qquad (11)$$

where $\delta_{nj} = 1$ for $n = j$, and zero otherwise. The structure of the matrix $W^{\ell}$ is illustrated in the Supplementary Information.

Applying the variation of constants formula to system (1) yields for $0 \le t_0 < t \le TL$:

$$x(t) = e^{-\alpha(t-t_0)} x(t_0) + \int_{t_0}^{t} e^{\alpha(s-t)} f(a(s)) \, \mathrm{d}s. \qquad (12)$$

In particular, for $t_0 = (\ell-1)T + (n-1)\theta$ and $t = (\ell-1)T + n\theta$ we obtain

$$x^{\ell}_n = e^{-\alpha\theta} x^{\ell}_{n-1} + \int_{t_0}^{t_0+\theta} e^{\alpha(s-(t_0+\theta))} f(a(s)) \, \mathrm{d}s, \qquad (13)$$

where $a(s)$ is given by (2). Note that the functions $\mathcal{M}_d(t)$, $b(t)$, and $J(t)$ are step functions that are constant on the integration interval. Approximating $x(s - \tau_d)$ by the value on the right $\theta$-grid point $x(t - \tau_d) \approx x((\ell-1)T + n\theta - n_d\theta)$ directly yields the network equation (6).

### Application to machine learning and a back-propagation algorithm.
We apply the system to two different types of machine learning tasks: image classification and image denoising. For the classification tasks, the size $P$ of the output layer equals the number of classes. We choose $f^{\mathrm{out}}$ to be the softmax function, i.e.,

$$\hat{y}_p = f^{\mathrm{out}}_p(\mathbf{a}^{\mathrm{out}}) = \frac{\exp(a^{\mathrm{out}}_p)}{\sum_{q=1}^{P} \exp(a^{\mathrm{out}}_q)}, \qquad p = 1, \dots, P. \qquad (14)$$

If the task is to denoise a greyscale image, the number of output nodes $P$ is the number of pixels of the image. In this case, clipping at the bounds 0 and 1 is a proper choice for $f^{\mathrm{out}}$, i.e.

$$\hat{y}_p = f^{\mathrm{out}}_p(\mathbf{a}^{\mathrm{out}}) = \begin{cases} 0, & \text{if } a^{\mathrm{out}}_p < 0, \\ a^{\mathrm{out}}_p, & \text{if } 0 \le a^{\mathrm{out}}_p \le 1, \\ 1, & \text{if } a^{\mathrm{out}}_p > 1. \end{cases} \qquad (15)$$

"Training the system" means finding a set of training parameters, denoted by the vector $\mathcal{W}$, which minimizes a given loss function $\mathcal{E}(\mathcal{W})$. Our training parameter vector $\mathcal{W}$ contains the input weights $w^{\mathrm{in}}_{nm}$, the non-zero hidden weights $w^{\ell}_{nj}$, and the

output weights $w_{pn}^{\text{out}}$. The loss function must be compatible with the problem type and with the output activation. For the classification task, we use the cross-entropy loss function

$$\mathcal{E}_{\text{CE}}(\mathcal{W}) := -\sum_{k=1}^{K}\sum_{p=1}^{P} y_p(k)\ln(\hat{y}_p(k)) = -\sum_{k=1}^{K}\ln(\hat{y}_{p_t(k)}(k)), \tag{16}$$

where $K$ is the number of examples used to calculate the loss and $p_t(k)$ is the target class of example $k$. For the denoising tasks, we use the rescaled mean squared error (MSE)

$$\mathcal{E}_{\text{MSE}}(\mathcal{W}) := \frac{1}{2K}\sum_{k=1}^{K}\sum_{p=1}^{P}\left(\hat{y}_p(k) - y_p(k)\right)^2. \tag{17}$$

We train the system by stochastic gradient descent, i.e., for a sequence of training examples $(\mathbf{u}(k), \mathbf{y}(k))$ we modify the training parameter iteratively by the rule

$$\mathcal{W}_{k+1} = \mathcal{W}_k - \eta(k)\nabla\mathcal{E}(\mathcal{W}_k, \mathbf{u}(k), \mathbf{y}(k)), \tag{18}$$

where $\eta(k) := \min(\eta_0, \eta_1/k)$ is a decreasing training rate.

If the node separation $\theta$ is sufficiently large, the local connections within the network become insignificant, and the gradient $\nabla\mathcal{E}(\mathcal{W})$ can be calculated using the classical back-propagation algorithm for multilayer perceptrons. Our numerical studies show that this works well if $\theta \geq 4$ for the considered examples. For smaller node separations, we need to take the emerging local connections into account. In the following, we first describe the classical algorithm, which can be used in the case of large $\theta$. Then we formulate the back-propagation algorithm for the Fit-DNN with significant local node couplings.

The classical back-propagation algorithm can be derived by considering a multilayer neural network as a composition of functions

$$\hat{\mathbf{y}} = f^{\text{out}}(\mathbf{a}^{\text{out}}(\mathbf{a}^L(\ldots(\mathbf{a}^1(\mathbf{a}^{\text{in}}(\mathbf{u})))))) \tag{19}$$

and applying the chain rule. The first part of the algorithm is to iteratively compute partial derivatives of the loss function $\mathcal{E}$ w.r.t. the node activations, the so called error signals, for the output layer

$$\delta_p^{\text{out}} := \frac{\partial\mathcal{E}(\mathbf{a}^{\text{out}})}{\partial a_p^{\text{out}}} = \hat{y}_p - y_p, \tag{20}$$

for $p = 1, \ldots, P$, and for the hidden layers

$$\delta_n^L := \frac{\partial\mathcal{E}(\mathbf{a}^L)}{\partial a_n^L} = f'(a_n^L)\sum_{p=1}^{P}\delta_p^{\text{out}}w_{pn}^{\text{out}}, \tag{21}$$

$$\delta_n^\ell := \frac{\partial\mathcal{E}(\mathbf{a}^\ell)}{\partial a_n^\ell} = f'(a_n^\ell)\sum_{i=1}^{N}\delta_i^{\ell+1}w_{in}^\ell, \quad \ell = L-1, \ldots, 1. \tag{22}$$

for $n = 1, \ldots, N$. Then, the partial derivatives of the loss function w.r.t. the training parameters can be calculated:

$$\frac{\partial\mathcal{E}(\mathcal{W})}{\partial w_{pn}^{\text{out}}} = \delta_p^{\text{out}}x_n^L, \tag{23}$$

for $n = 1, \ldots, N+1$ and $p = 1, \ldots, P$,

$$\frac{\partial\mathcal{E}(\mathcal{W})}{\partial w_{nj}^\ell} = \delta_n^\ell x_j^{\ell-1}, \tag{24}$$

for $\ell = 2, \ldots, L$, $j = 1, \ldots, N+1$ and $n = 1, \ldots, N$, and

$$\frac{\partial\mathcal{E}(\mathcal{W})}{\partial w_{nm}^{\text{in}}} = \delta_n^1 g'(a_n^{\text{in}})u_m, \tag{25}$$

for $m = 1, \ldots, M+1$ and $n = 1, \ldots, N$. For details, see ref. [54] or ref. [3].

Taking into account the additional linear connections, we need to change the way we calculate the error signals $\delta_n^\ell$ for the hidden layers. Strictly speaking, we cannot consider the loss $\mathcal{E}$ as a function of the activation vector $\mathbf{a}^\ell$, for $\ell = 1, \ldots, L$, because there are connections skipping these vectors. Also, Eq. (19) becomes invalid. Moreover, nodes of the same layer are connected to each other. However, the network has still a pure feed-forward structure, and hence, we can apply back-propagation to calculate the error signals node by node. We obtain the following algorithm to compute the gradient.

*Step 1*: Compute

$$\delta_p^{\text{out}} := \frac{\partial\mathcal{E}}{\partial a_p^{\text{out}}} = \hat{y}_p - y_p, \tag{26}$$

for $p = 1, \ldots, P$.

*Step 2*: Let $\Phi := \alpha^{-1}(1 - e^{-\alpha\theta})$. Compute the error derivatives w.r.t. the node states of the last hidden layer

$$\Delta_N^L := \frac{\partial\mathcal{E}}{\partial x_N^L} = \sum_{p=1}^{P}\delta_p^{\text{out}}w_{pN}^{\text{out}}, \tag{27}$$

and

$$\Delta_n^L := \frac{\partial\mathcal{E}}{\partial x_n^L} = \Delta_{n+1}^L e^{-\alpha\theta} + \sum_{p=1}^{P}\delta_p^{\text{out}}w_{pn}^{\text{out}}, \tag{28}$$

for $n = N-1, \ldots, 1$. Then compute the error derivatives w.r.t. the node activations

$$\delta_n^L := \frac{\partial\mathcal{E}}{\partial a_n^L} = \Delta_n^L\Phi f'(a_n^L), \tag{29}$$

for $n = 1, \ldots, N$.

*Step 3*: Repeat the same calculations as in step 2 iteratively for the remaining hidden layers $\ell = L-1, \ldots, 1$, while keeping the connection between the nodes $x_N^\ell$ and $x_1^{\ell+1}$ in mind. That is, compute

$$\Delta_N^\ell := \frac{\partial\mathcal{E}}{\partial x_N^\ell} = \Delta_1^{\ell+1}e^{-\alpha\theta} + \sum_{i=1}^{N}\delta_i^{\ell+1}w_{iN}^{\ell+1}, \tag{30}$$

and

$$\Delta_n^\ell := \frac{\partial\mathcal{E}}{\partial x_n^\ell} = \Delta_{n+1}^\ell e^{-\alpha\theta} + \sum_{i=1}^{N}\delta_i^{\ell+1}w_{in}^{\ell+1}, \tag{31}$$

for $n = N-1, \ldots, 1$. Computing the error derivatives w.r.t. the node activations works exactly as for the last hidden layer:

$$\delta_n^\ell := \frac{\partial\mathcal{E}}{\partial a_n^\ell} = \Delta_n^\ell\Phi f'(a_n^\ell), \tag{32}$$

for $n = 1, \ldots, N$.

*Step 4*: Calculate weight gradient using Eqs. (23)–(25).

The above formulas can be derived by the chain rule. Note that many of the weights contained in the sums in Eq. (30) and Eq. (31) are zero when the weight matrices for the hidden layers are sparse. In this case, one can exploit the fact that the non-zero weights are arranged on diagonals and rewrite the sums accordingly to accelerate the computation

$$\sum_{i=1}^{N}\delta_i^{\ell+1}w_{in}^{\ell+1} = \sum_{\substack{d=1 \\ 1 \leq n+n_d' \leq N}}^{D}\delta_{n+n_d'}^{\ell+1}v_{d,n+n_d'}^{\ell+1} \tag{33}$$

For details we refer to the Supplementary Information. Additionally, the training process is illustrated in Supplementary Movie 1.

**Data augmentation, input processing, and initialization**. For all classification tasks, we performed an augmentation of the training input data by adding a small Gaussian noise to the images and by pixel jittering, i.e., randomly shifting the images by at most one pixel horizontally, vertically, or diagonally. For the CIFAR-10/100 tasks, we also applied a random rotation of maximal $\pm 15°$ and a random horizontal flip with the probability 0.5 to the training input images. Further, we used dropout[55] with a dropout rate of 1% for the CIFAR-10/100 tasks. For the denoising task, we performed no data augmentation.

Moreover, for the five classification tasks, we used the input preprocessing function $g(a) = \tanh(a)$. For the denoising task, we applied no nonlinear input preprocessing, i.e., $g(a) = a$. The weights were always initialized by Xavier initialization[56]. In all cases, we used 100 training epochs.

## Data availability

The MNIST data[40] (labeled images of handwritten digits) used in this study are available in the MNIST database: http://yann.lecun.com/exdb/mnist/. The Fashion-MNIST data[41] (labeled images of fashion articles) used in this study are available in the Fashion-MNIST database: https://github.com/zalandoresearch/fashion-mnist, MIT License © 2017 Zalando SE. The CIFAR-10 and CIFAR-100 data[42] (labeled images of objects, animals, plants, or people) used in this study are available in the CIFAR-10 and CIFAR-100 databases: https://www.cs.toronto.edu/~kriz/cifar.html. The cropped SVHN data[43] (labeled images of house numbers extracted from Google Street View photos) used in this study are available in the SVHN database: http://ufldl.stanford.edu/housenumbers/.

## Code availability

The source code[57] to reproduce the results of this study is freely available on GitHub: https://github.com/flori-stelzer/deep-learning-delay-system/tree/v1.0.0.

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

## Acknowledgements

F.S. and S.Y. acknowledge funding by the "Deutsche Forschungsgemeinschaft" (DFG) in the framework of the project 411803875 and IRTG 1740. A.R. and I.F. acknowledge support by the Spanish State Research Agency, through the Severo Ochoa and María de Maeztu Program for Centers and Units of Excellence in R&D (MDM-2017-0711). R.V. thanks the financial support from the Estonian Centre of Excellence in IT (EXCITE) funded by the European Regional Development Fund, through the research grant TK148. R.V. thanks the financial support from the EU H2020 program via the TRUST-AI (952060) project.

## Author contributions

All authors (F.S., A.R., R.V., I.F. and S.Y.) contributed extensively to the work presented in this paper and to the writing of the manuscript.

## Funding

## Competing interests

The authors declare no competing interests.
