## [Peer Review File · Nature Communications]

Reviewers' Comments:

Reviewer #1:

Remarks to the Author:

This paper proposes a new mathematical model which can emulate a complete multilayer perceptron using just one neuron and feedback-modulated delay loops in time. In essence, it is a theoretical paper which may have far reaching implications in neuromorphic computing if the idea will be adopted and validated by the community and if vast amount of research will be put into it further.

However, in the actual form, there are a number of reasons for which I believe that the paper is not ready yet for publication. To address them, I have several suggestions and questions to the authors which may help improving the paper overall quality for the next revision.

1) What keep me to say that the paper is visionary is Ref [13] which actually introduced the idea for, true, a much more simpler variant of neural networks. To clarify better the paper contributions, I would suggest stating them clearly somewhere in the first part of the paper, mentioning clearly what is new in this paper in comparison with Ref [13] and the large body of work done on the topic in meanwhile.

2) To put the theoretical model in hardware is not a trivial task. I am not an electrical or a photonics engineer, but I believe that the paragraph starting with "The Fit-DNN approach..." (line 252) discusses the possibility of putting the proposed model into hardware with too much ease giving the wrong impression to the reader that it is a relatively easy task. I believe that a more realistic discussion has to be made here, and the real engineering and research challenges which have to be solved in order to have a working hardware prototype have to be discussed seriously. Perhaps, I am wrong, but to orchestrate all the delay loops just with photonics may be very challenging, while converting from optical signal to electrical signal may be too slow. Wouldn't be easier to have a first prototype in silicon? Also, from a hardware design perspective, how easy will scale such a device to millions of delay loops to reach the scale of current neural network size (an advantage here is, I assume, the linear relation between the number of delay loops and neurons)?

3) As the authors mentioned, emulating sparse neural networks may be indeed a commonsense approach for the proposed model. Still, in Ref [37] and in all of the follow up sparse training works, it has been shown that just the dynamic sparsity (optimizing and adapting the sparse connectivity while training) can outperform the dense network constantly. Is it easy to incorporate the idea of dynamic sparsity into the proposed theoretical model?

4) Just a curiosity, why are using the sine function (an unusual choice for neural networks) and not sigmoid?

5) In terms of evaluation, I assume that it can be quite time consuming to train a model with an increased number of hidden neurons even if you wrote the code in C++. Still, for a better overview of the model performance a comparison with the equivalent MLP model (with exactly the same static sparse connectivity, number of neurons, and any other setting) would help in understanding better the model performance. Also, a fully connected MLP (same settings and neurons) would be interesting to see.

6) Minor, denoising is a different task from classification which uses a different evaluation metric. Why are you presenting them together in the same table? It can confuse the reader.

7) Is it possible to perform experiments also on a more challenging dataset (e.g. CIFAR 100)?

8) [optional] How difficult would be to adapt this idea to recurrent neural networks and convolutional neural networks?

9) Please perform a careful proof-read of the whole paper (including the mathematical details) as not everything is crystal clear and there are still some typos (e.g. "???" in supplementary materials)

10) Very nice video.

Reviewer #2:

Remarks to the Author:

This manuscript describes an innovative method of using a dynamical system with time delay to replicate an artificial neural network with several layers. The key aspect is that only a single nonlinearity is required. Instead of the usual weight matrices, the authors use a number of delay couplings, which are suitably modulated in time to encode the weight matrix elements. One attractive feature of the proposed concept is that it might at least in principle be realised using optical means, for example with optical fibres and lasers. This might give rise to innovative new hardware realisations of this concept.

While I do acknowledge and appreciate the innovative content of the presented material, I also think that there are obvious fundamental weaknesses of the proposed method, which the authors neglect to discuss. As it stands, this could lead to a misleading impression on the potential for this method, and I would request that the authors address the two most obvious issues.

The first issue relates to the obtainable performance. One should recall that the recent success of Deep Learning is strongly connected to the efficiency with which modern hardware can run many thousands operations in parallel. In this regard, the proposed concept of using only a single nonlinearity through which all operations need to be funnelled looks like a fundamental bottleneck in the design. This bottleneck is not properly discussed in the manuscript and only implicitly alluded to in section 4.2 when the case of small theta is discussed. While the authors rightly claim that the design is highly flexible, it should also be made clear that the design will not scale well for a large number of nodes due to the performance bottleneck.

The second issue concerns the claim of "Deep Learning with a Single Neuron" in the title. I do believe that this is misleading, as at least in my understanding the single neuron does not do the training at all. As explained in 4.3 the training is done using a standard computer with classical (or modified) backpropagation. The fact that the backpropagation is modified for small theta to account for the cross-talk does not mean that the learning happens using the single neuron. Therefore, even if the method proposed by the authors was realised in hardware, this hardware would be of no help for the training. Since training is commonly the much more difficult task, it should be made clear throughout the manuscript that the single neuron will only perform the task of the forward propagation. I would also suggest to change the title to avoid this impression.

In my opinion, the authors should address the two mentioned issues satisfactorily before publication can be recommended.

Reviewer #3:

Remarks to the Author:

The article entitled "Deep Learning with a Single Neuron: Folding a Deep Neural Network in Time using Feedback-Modulated Delay Loops" from Stelzer et al. proposes a new approach for high-performance machine learning. Their method, that they call "Folded-in-time DNN" (Fit-DNN), is based in the folding of an arbitrary size DNN into a single nonlinear neuron. The dynamical state of network is thereby translated into the time domain of that neuron, and modulation feedback is used to adjust the connection weights. The method is successfully tested against various machine learning benchmarks. The data analysis and interpretation is sound and enough information is provided to permit the replication of the obtained results.

The article is quite well written, informative and complete. It is in my opinion a valuable contribution to the field, that has a level of impact and novelty that warrants publication in Nature Communications. I would however invite the authors to address the three following points:

- Equations (1) and (2) indicate that the neuron obeys an Ikeda-like equation. Is the approach

proposed by the authors valid with other types of time-delayed systems?

- The proposed method seems at the intersection of "conventional" machine learning (use of backpropagation, etc.) and delay-based reservoir computing (single neuron, etc.). The reader would intuitively assume that the new method intends to leverage from the best of both worlds. A dedicated section where the performance of the Fit-DNN would be discussed in contrast to the two other approaches would be of interest.

- Could the authors elaborate on the efficiency of their method from the viewpoint of computation time?

Detailed Response to the referee reports:

Summary of Main Changes

1. We have extended our numerical studies by a classification task considering the 20 superclasses of the CIFAR-100 dataset (Sec. 2.5, Tables 1 and 2).
2. We have improved the back-propagation algorithm for Fit-DNN so that the computational time scales linearly with the number of nodes. (Sec. “Methods” and Sec. 3 of Supplemental Material have been significantly rewritten).
3. We have changed the title to avoid confusion in the phrase “Deep Learning with a Single Neuron”.
4. We have added an explanation for the choice of the Ikeda-like system. We also mentioned that there are no fundamental obstacles for using other types of time-delayed systems (Sec. 2.1).
5. We have addressed the dynamic sparsity idea (Sec. 2.4, 3.3, Discussion).
6. We have emphasized more clearly the novel aspects in comparison to Ref. [13] (Introduction).
7. We have discussed the model performance in comparison with the equivalent MLP model (Sec. 3.3).
8. We have confirmed and discussed the possibility of implementing folded in time recurrent neural networks (Sec. 3.1).
9. We have added to the discussion of the strengths and weaknesses of our method based on the questions and points of criticism of the reviewers (Sec. 3.5).
10. We have extended the discussion of potential hardware implementation (Sec. 3.4).

We also provide a “diff” version of our revised manuscript in which deletions from the original manuscript are printed in red, and additions are printed in blue. In the following response, quotations of additions (deletions) to the revised manuscript are printed in blue (red).

Response to report of Referee #1

This paper proposes a new mathematical model which can emulate a complete multilayer perceptron using just one neuron and feedback-modulated delay loops in time. In essence, it is a theoretical paper which may have far reaching implications in neuromorphic computing if the idea will be adopted and validated by the community and if vast amount of research will be put into it further.

However, in the actual form, there are a number of reasons for which I believe that the paper is not ready yet for publication. To address them, I have several suggestions and questions to the authors which may help improving the paper overall quality for the next revision.

We thank the reviewer for the positive evaluation of the manuscript and for pointing out its possible far-reaching implications.

1) What keep me to say that the paper is visionary is Ref [13] which actually introduced the idea for, true, a much more simpler variant of neural networks. To clarify better the paper contributions, I would suggest stating them clearly somewhere in the first part of the paper, mentioning clearly what is new in this paper in comparison with Ref [13] and the large body of work done on the topic in meanwhile.

Our manuscript has taken some inspiration from Ref. [13], in which the concept of time-delay reservoir computing was introduced. Ref. [13] employs a system with a single delay loop to implement a simple recurrent neural network with fixed internal weights. Compared to Ref. [13], there are several new aspects in our work.

1. We consider multiple modulated delay loops that allow a more flexible network connection structure, and even allow for fully connected layers. The possibility of using delay systems to emulate networks with more flexible connection structures have been studied previously in [45,46], however, in a different manner to our work and restricted to the case of fixed equal weights.
2. Our method introduces the ability to implement, in principle, arbitrary connectivity with adaptable weights. It is the temporal modulation of the feedback signals which enables adaptation.
3. We have demonstrated that the weights can be effectively trained by gradient descent training. This allows us to use a delay system to implement neural networks with trainable hidden layers.

To clarify the comparison with Ref. [13] and other recent research, we modified the last paragraph in the introduction:

Our concept of folded-in-time deep neural networks also benefits from time-multiplexing, but uses it in a more intricate manner going conceptually beyond by allowing for the implementation

of multi-layer feed-forward neural networks with adaptable hidden layer connections and, in particular, the applicability of the gradient descent method for their training. We present the Fit-DNN concept and show its versatility and applicability by solving benchmark tasks.

2) To put the theoretical model in hardware is not a trivial task. I am not an electrical or a photonics engineer, but I believe that the paragraph starting with “The Fit-DNN approach...” (line 252) discusses the possibility of putting the proposed model into hardware with too much ease giving the wrong impression to the reader that it is a relatively easy task. I believe that a more realistic discussion has to be made here, and the real engineering and research challenges which have to be solved in order to have a working hardware prototype have to be discussed seriously. Perhaps, I am wrong, but to orchestrate all the delay loops just with photonics may be very challenging, while converting from optical signal to electrical signal may be too slow.

Wouldn't be easier to have a first prototype in silicon? Also, from a hardware design perspective, how easy will scale such a device to millions of delay loops to reach the scale of current neural network size (an advantage here is, I assume, the linear relation between the number of delay loops and neurons)?

The appeal of our concept does not rely on its possibility of a hardware implementation. Still, the latter represents an interesting perspective. Moreover, discussing potential analog hardware implementations, our concept is not linked to particular hardware, and we do not want to limit it to a certain platform and by that lose a significant part of the readership. But exemplarily, we mentioned in the text that a suitable candidate could be a photonic neuromorphic implementation. This is motivated by the successful reports of delay-based photonic reservoir computing implementations [13-19]. But we emphasize that delay-based reservoir computing has also been successfully implemented using electronic systems, magnetic spin systems, MEMS, acoustic and other platforms. Certainly, realizations on different hardware platforms come with different challenges. In the following, we exemplify the requirements for a photonic (optoelectronic) scheme and discuss how realistic their implementation is. In fact, such an implementation requires only a light source, $1:(2^N-1)$ fiber couplers and optical fibers with different lengths. Those are cheap and easy to obtain components. Moreover, for every delay loop, a modulator is needed, which can be implemented via Mach-Zehnder intensity modulators. Finally, fast photodetectors (one for all delay loops and one for the output) would be required, as well as an optical amplifier or an electrical amplifier which could be used to compensate for roundtrip losses. Those are all standard components from telecommunications which can be obtained for low costs with GHz bandwidths (10s to 100s of Euros per detector and modulator), and can even easily reach bandwidths of several tens of GHz for some higher costs (thousands of Euros per detector or modulator). Moreover, all these components can, in principle, be integrated into Photonic Integrated circuits. An integration would only reduce the flexibility in the choice of delay times and desired connectivity.

Conversion from optical signals to electrical signals can be done extremely fast (faster than the clock rate of today's fast electronic processors) and only two photodetectors, independent of the number of virtual nodes and number of delay loops, are needed. In the lab of one of the authors,

modulation capabilities up to 96 GSamples per second and acquisition with 40GHz analog bandwidth and 256GSa/s are available, providing the perspective for a photonic implementation. In contrast, an electronic implementation in silico would come with some particular advantages, but would, e.g., suffer from the fact that delay lines are not that easily and efficiently to be implemented, while in photonic it requires just a piece of standard telecommunication fiber (or a passive optical waveguide in an integrated version).

In order to get to DNN with many hidden layers, amplification might be required to compensate for the losses from layer to layer. In the photonic scheme, this would be implemented via a semiconductor optical amplifier for fiber-optical implementations, or gain sections in photonic integrated circuits. Also those types of functional elements are standard technology.

Millions of delay loops are not needed, since the number of delay loops scales only linearly with the number of the virtual nodes of a single layer. For sparse connectivity, the number of delay loops can even be much lower than the number of the virtual nodes of a single layer. While scaling to the sizes of current digitally simulated neural networks would be quite challenging, our proposed approach could in principle require orders of magnitude fewer components than a direct hardware implementation, where each artificial neuron and weight would need to be represented by its own element. An aspect that also should not be forgotten, is the fact that through using the same physical element to emulate all the neurons, we essentially guarantee that the functional neurons in the equivalent network are identical. Parameter mismatches and non-identical elements can be a problem in implementations relying on large arrays of hardware neurons.

In comparison to other analog hardware or neuromorphic platforms of conventional DNN, it is worth noting that those approaches require a much larger number of precision-fabricated and tuned elements. They might not depend on the time-unfolding, giving the impression that they would be much faster. However, in reality, due to the use of much fewer but much faster components, our approach might turn out very competitive, in terms of speed, cost, and energy consumption.

Altogether, although we are providing some perspectives for hardware implementations here, we emphasize once more that this is a manuscript introducing and discussing a novel concept to realize DNN. A particular hardware implementation discussed in this manuscript would not only restrict the perspective but would also go far beyond the scope of this work. A hardware implementation would require a demanding and very detailed and specific characterization of its properties, repelling a large part of the targeted audience and diluting the general aspects of our novel approach.

Nevertheless, we have extended the discussion of hardware implementation in subsection 3.4 of the “Discussion” section:

In addition to providing a dynamical systems perspective on DNNs, Fit-DNNs can also serve as blueprints for specialized DNN hardware. The Fit-DNN approach is agnostic concerning the type of nonlinearity, enabling flexibility of implementations. A suitable candidate could be a photonic neuromorphic implementation [13, 14, 15, 16, 52, 53, 20], where a fast artificial neuron can be realized with the Gigahertz timescale range. Photonic systems have already been used to

construct delay-based reservoir computers. In retrospect, it is quite clear how instrumental the reduced hardware requirement of a delay-based approach was in stimulating the current ecosystem of reservoir computing implementations. For example, the delay-based reservoir computing has been successfully implemented using electronic systems, magnetic spin systems, MEMS, acoustic, and other platforms. We hope that for the much larger community around DNNs, a similarly stimulating effect can be achieved with the Fit-DNN approach we presented here, since it also drastically reduces the cost and complexity for hardware-based DNNs.

Certainly, realizations on different hardware platforms face different challenges. In the following, we exemplify the requirements for a photonic (optoelectronic) scheme. Such an implementation requires only one light source, a few fiber couplers, and optical fibers of different lengths. The modulations of the delay loops can be implemented using Mach-Zehnder intensity modulators. Finally, only two fast photodetectors (one for all delay loops and one for the output) would be required, as well as an optical amplifier or an electrical amplifier which could be used to compensate for roundtrip losses. Those are all standard telecommunication components. The conversion from optical to electrical signals can be done extremely fast, faster than the clock rate of today's fast electronic processors, and only two photodetectors are needed, regardless of the number of virtual nodes and number of delay loops.

3) As the authors mentioned, emulating sparse neural networks may be indeed a commonsense approach for the proposed model. Still, in Ref [37] and in all of the follow up sparse training works, it has been shown that just the dynamic sparsity (optimizing and adapting the sparse connectivity while training) can outperform the dense network constantly. Is it easy to incorporate the idea of dynamic sparsity into the proposed theoretical model?

In our approach, removing or adding a delay loop would change an entire diagonal in the hidden weight matrices. This is due to the Fit-DNN's special sparsity pattern: the delay-induced connections are arranged on diagonals of the connection matrices. The sparsity training algorithms in [37] and related references rely on the ability to flexibly choose the non-zero connections of sparse networks. Therefore, they are not directly applicable to the Fit-DNN. We tested pruning and further dynamic sparsity training methods for our model. We observed that removing all weights of a diagonal at the same time perturbed the previous training too much so that the method fails. Also, adding diagonals incrementally while testing which positions were the most favorable did not improve the performance.

Nevertheless, we expect that it is possible to find a suitable method to optimize the choice of delays. One candidate for such a method is pruning by slowly fading out diagonals that contain weaker connections on average. However, further investigations of specific sparsity training methods for the Fit-DNN are beyond the scope of this manuscript.

To address the sparsity issue and to discuss our previous attempts to adapt existing sparsity training methods for the Fit-DNN, we added the following paragraph in the Discussion section:

It has been shown that dynamic sparsity [38, 39] can outperform dense networks and, fundamentally, Fit-DNNs are intrinsically compatible with certain kinds of sparsity. However, in

our approach, removing or adding a delay loop would change an entire diagonal in the hidden weight matrices. Therefore, sparsity training algorithms such as [38, 39] and related works are not directly applicable to the Fit-DNN. Our preliminary tests have shown that removing the weights of a diagonal at the same time disturbs the previous training too much, so the method fails. Nevertheless, we expect that it is possible to find a suitable method to optimize the choice of delays. Therefore, further investigation of specific sparsity training methods for the Fit-DNN would be very welcome. One candidate for such a method could be pruning by slowly fading diagonals that contain weaker connections on average.

4) Just a curiosity, why are using the sine function (an unusual choice for neural networks) and not sigmoid?

Since the sine nonlinearity is compatible with an optoelectronic implementation (see, e.g., [14,20,33]), we considered it as the default choice for the numerics. However, we have also performed tests with other, more traditional nonlinearities such as “tanh” and “ReLU”; see Table 2 in the Supplementary Information. The Fit-DNN works equally well for all of these nonlinearities, and we conclude that the particular type of nonlinearity does not seem to be crucial.

5) In terms of evaluation, I assume that it can be quite time consuming to train a model with an increased number of hidden neurons even if you wrote the code in C++. Still, for a better overview of the model performance a comparison with the equivalent MLP model (with exactly the same static sparse connectivity, number of neurons, and any other setting) would help in understanding better the model performance. Also, a fully connected MLP (same settings and neurons) would be interesting to see.

The training is indeed time-consuming due to the fact that we need to solve the delay-dynamical equation for each training step. (This could be overcome if “online” training, i.e., training on a hardware-implemented system, can be realized.) The following table compares the training time for the Fit-DNN with the training time of the corresponding MLP (for MNIST, 10 epochs, with the standard parameters listed in Table 1 of the manuscript, on Intel Core i7-10700):

CPU time in seconds	Fit-DNN (sparse D=100)	MLP (sparse D=100)	Fit-DNN (full D=199)	MLP (full D=199)
Total	689	312	958	307
Forward propagation or DDE resp.	529	151	787	145
Backpropagation	81	81	88	86
other	79	80	83	76

Note that we used a straightforward implementation of the MLP and backpropagation algorithm for comparison and that there is certainly room for optimization (e.g., parallelization). Moreover, the required time for solving the DDE strongly depends on the chosen numerical integration method and step size. We were using the Heun method with a step size of $h = \theta/32$ for $\theta < 2$ and $h = 1/16$ for $\theta \geq 2$, i.e., the CPU times in the above table were measured with $h = 1/64$.

The strength of our approach is so far the potentially faster inference. We see a potential that our methods can achieve fast inference times when it is realized in hardware. The training process is indeed rather time-consuming because we need to solve the DDE for the forward propagation. One option to overcome this limitation would be to perform the forward propagation even during the training process on specialized hardware. However, that would certainly be more difficult to implement than “offline” training on a conventional computer. To emphasize this, we included the following passage in the manuscript (Sec. 3.5):

We would also like to point out that the potential use of very fast hardware components is accompanied by a possibility of fast inference. However, a fast hardware implementation of the Fit-DNN will not accelerate the training process, because a traditional computer is still required, at least for the back-propagation of errors. If the forward propagation part of the training process is also performed on a traditional computer, the delay equation must be solved numerically for each training step, leading to a significant increase in training time. Therefore, the presented method is most suitable when fast inference and/or high hardware efficiency are prioritized. We would like to point out that the integration of the training process into the hardware-part could be addressed in future extensions of our concept.

Moreover, the computational costs for the back-propagation, as reported in the first submission, scale quadratically with the number of nodes N per hidden layer, i.e., it is slow for “wide” networks. That is, your assumption is correct if this version of back-propagation is applied. We have taken your comment as an opportunity to improve the back-propagation algorithm for the Fit-DNN. We found that it is actually possible to reformulate the algorithm such that the computation time scales only linearly with N . This can either be achieved by rearranging the formulas (for steps 2 and 3) or by directly alternating the derivation of the algorithm.

Accordingly, we performed a comprehensive revision of the Methods Section and Section 3 of the Supplementary Information.

The updated version of the back-propagation algorithm for the Fit-DNN follows the traditional back-propagation scheme for forward-connected networks, with the minor difference that we have to distinguish between linear and nonlinear connections entering the same node. In terms of computational cost, the updated algorithm is equivalent to back-propagation for the traditional MLP. This is also clearly shown by the measured CPU times in the table above. Thus, the computational time for the back-propagation is much smaller than the time required to solve the DDE. Since the latter strongly depends on the choice and details of the numerical integration method, no general statement can be made about how much longer the training takes compared to an MLP. For our default parameters and the MNIST task, training the Fit-DNN takes about three times longer than training an equivalent MLP.

For a comparison of the Fit-DNN's performance in terms of accuracy to an MLP, we refer to Fig. 4 of the manuscript. Here we show the accuracy that we can achieve with a Fit-DNN with different node separations for four different classification tasks. The accuracy of an equivalent MLP is indicated by a black line for each task. In order to clarify this, we updated the caption of Fig. 4 as follows: we changed the sentence:

The accuracy obtained in the map limit case $\theta \rightarrow \infty$ is shown by the horizontal black line.

to
The accuracy obtained in the map limit case $\theta \rightarrow \infty$ is shown by the horizontal black line (this corresponds to the classical sparse multilayer perceptron).

Moreover, note that our accuracy results were obtained with a moderate number of trainable parameters. Our standard setup achieved an accuracy of 98.49 % for MNIST using approximately 84000 training parameters and an accuracy of 51.42 % for CIFAR using approximately 318000 training parameters. For comparison: to our knowledge, for MLPs, the record accuracy for the MNIST classification task is 99.65 % and was achieved with 11965000 training parameters [Dan Claudiu Ciresan, Ueli Meier, Luca Maria Gambardella, Juergen Schmidhuber: "Deep Big Simple Neural Nets Excel on Handwritten Digit Recognition" (2010)]. For CIFAR-10, accuracies of 56.84 % (ReLU and linear activation) and 65.76 % ("zero-bias" and linear activation) have been achieved with MLPs with 20328000 training parameters. By extensive data augmentation, the accuracy could be increased to 78.62 % [Zhouhan Lin, Roland Memisevic, Kishore Konda: "How far can we go without convolution: Improving fully-connected networks" (2015)]. For higher success rates other methods than MLPs are required.

6) Minor, denoising is a different task from classification which uses a different evaluation metric. Why are you presenting them together in the same table? It can confuse the reader.

We have restructured Table 2 of the main text and Tables 1 and 2 of the Supplementary Information to avoid confusion.

7) Is it possible to perform experiments also on a more challenging dataset (e.g. CIFAR 100)?

To address this question, we extended our numerical studies by a classification task considering the 20 superclasses of the CIFAR-100 dataset.

As a result, we have introduced the following changes:

- In Sec. 2.5 (Benchmark tasks):

To demonstrate the computational capabilities of the Fit-DNN over these regimes, we considered five image classification tasks: MNIST [40], Fashion-MNIST [40], CIFAR-10, CIFAR-100 considering the coarse class labels [42], and the cropped version of SVHN [43].

- In Table 1, the new row (c) is added and the caption is modified related to the new CIFAR100 tests.
- A few other minor changes.

8) [optional] How difficult would be to adapt this idea to recurrent neural networks and convolutional neural networks?

It is indeed possible to adapt this idea to recurrent neural networks (RNNs). For this, we only need to use T-periodic modulation functions for the delayed feedback terms. In this way, we obtain hidden layers with shared weights and thus an RNN.

In other words, it is always possible to implement an RNN when the implementation of very deep DNNs with identical layers is possible. Therefore, we expect our system to be well-suited for RNN implementations. Moreover, a certain type of RNN has already been presented in Ref. [13], but without adaptable weights and only N internal connections for the hidden layers.

Implementing convolutional neural networks (CNNs) with the folded-in-time approach is much more challenging. The reason is that pooling layers are not compatible with the type of nodes that appear in Fit-DNN setup. One option would be to try more complicated dynamics, such as systems with distributed delays. In our opinion, extending the folded-in-time approach to CNNs would be a significant and challenging step to take this approach further.

We have modified one sentence in the Discussion as follows:

For instance, one can implement different layer sizes, multiple nonlinear elements, and combine different structures such as recurrent neural networks with trainable hidden layers.

9) Please perform a careful proof-read of the whole paper (including the mathematical details) as not everything is crystal clear and there are still some typos (e.g. “??” in supplementary materials)

Done.

10) Very nice video.

Thank you very much for the positive comments.

Response to report of Referee #2

This manuscript describes an innovative method of using a dynamical system with time delay to replicate an artificial neural network with several layers. The key aspect is that only a single nonlinearity is required. Instead of the usual weight matrices, the authors use a number of delay couplings, which are suitably modulated in time to encode the weight matrix elements. One attractive feature of the proposed concept is that it might at least in principle be realised using optical means, for example with optical fibres and lasers. This might give rise to innovative new hardware realisations of this concept.

Thank you for the positive evaluation of the manuscript and for pointing out its innovative nature.

While I do acknowledge and appreciate the innovative content of the presented material, I also think that there are obvious fundamental weaknesses of the proposed method, which the authors neglect to discuss. As it stands, this could lead to a misleading impression on the potential for this method, and I would request that the authors address the two most obvious issues.

Thank you for pointing out the insufficient discussion of the possible weaknesses of our method. We take this comment very seriously. Below are the responses and the list of corresponding changes in the manuscript. We also created a new subsection 3.5 (Trade-Offs) to address the limitations of the method.

The first issue relates to the obtainable performance. One should recall that the recent success of Deep Learning is strongly connected to the efficiency with which modern hardware can run many thousands operations in parallel. In this regard, the proposed concept of using only a single nonlinearity through which all operations need to be funnelled looks like a fundamental bottleneck in the design. This bottleneck is not properly discussed in the manuscript and only implicitly alluded to in section 4.2 when the case of small theta is discussed. While the authors rightly claim that the design is highly flexible, it should also be made clear that the design will not scale well for a large number of nodes due to the performance bottleneck.

The nodes of the Fit-DNN are indeed processed sequentially, i.e., the processing time of all hidden layers of the Fit-DNN is θ multiplied by the total number of nodes. The following modified paragraph (sec. Discussion) discusses this problem and proposes possible ways to address it:

Since only one nonlinear node and one fast read-out element are absolutely necessary in our approach, ultrafast components could be used that would be unrealistic or too expensive for full DNN implementations. At the same time, since the single nonlinear element performs all nonlinear operations sequentially with node separation θ , parallelization cannot be applied in this approach. The overall processing time scales linearly with the total number of nodes LN and

with the node separation θ . Possible ways to address this property that could represent a limitation in certain applications include the use of a small node separation θ [13] or multiple parallel copies of Fit-DNNs. In this way, a tradeoff between the number of required hardware components and the amount of parallel processing is possible. At the same time, the use of a single nonlinear node comes with the advantage of almost perfect homogeneity of all folded nodes, since they are realised by the same element.

The second issue concerns the claim of "Deep Learning with a Single Neuron" in the title. I do believe that this is misleading, as at least in my understanding the single neuron does not do the training at all. As explained in 4.3 the training is done using a standard computer with classical (or modified) backpropagation. The fact that the backpropagation is modified for small theta to account for the cross-talk does not mean that the learning happens using the single neuron. Therefore, even if the method proposed by the authors was realised in hardware, this hardware would be of no help for the training. Since training is commonly the much more difficult task, it should be made clear throughout the manuscript that the single neuron will only perform the task of the forward propagation. I would also suggest to change the title to avoid this impression. In my opinion, the authors should address the two mentioned issues satisfactorily before publication can be recommended.

From the comment, we see that the chosen wording "Deep Learning with a Single Neuron" may be misleading. As an alternative title, we suggest:

Deep Neural Networks using a Single Neuron: Folded-in-Time Architecture using Feedback-Modulated Delay Loops

We have also found a place in the Discussion section, where a similar misleading may occur. The sentence

As a result, just one neuron is sufficient to fold the whole complexity of the network architecture. is changed to

As a result, just one neuron with feedback is sufficient to fold the entire complexity of the network.

Additionally, we mention that a possible hardware implementation would not help in the training process. We added the following passage:

We would also like to point out that the potential use of very fast hardware components is accompanied by a possibility of fast inference. However, a fast hardware implementation of the Fit-DNN will not accelerate the training process, because a traditional computer is still required, at least for the back-propagation of errors. If the forward propagation part of the training process is also performed on a traditional computer, the delay equation must be solved numerically for each training step, leading to a significant increase in training time. Therefore, the presented method is most suitable when fast inference and/or high hardware efficiency are prioritized. We would like to point out that the integration of the training process into the hardware-part could be addressed in future extensions of our concept.

Response to report of Referee #3

The article entitled “Deep Learning with a Single Neuron: Folding a Deep Neural Network in Time using Feedback-Modulated Delay Loops” from Stelzer et al. proposes a new approach for high-performance machine learning. Their method, that they call “Folded-in-time DNN” (Fit-DNN), is based in the folding of an arbitrary size DNN into a single nonlinear neuron. The dynamical state of network is thereby translated into the time domain of that neuron, and modulation feedback is used to adjust the connection weights. The method is successfully tested against various machine learning benchmarks. The data analysis and interpretation is sound and enough information is provided to permit the replication of the obtained results.

The article is quite well written, informative and complete. It is in my opinion a valuable contribution to the field, that has a level of impact and novelty that warrants publication in Nature Communications. I would however invite the authors to address the three following points:

Thank you for the positive evaluation of the manuscript and for mentioning its suitability for Nature Communications.

- Equations (1) and (2) indicate that the neuron obeys an Ikeda-like equation. Is the approach proposed by the authors valid with other types of time-delayed systems?

There are several reasons for choosing the Ikeda-like systems. It is a convenient system for delay-based reservoir computing, which has been introduced in [13]. A simple structure of the instantaneous terms – $\alpha x(t)$ allows us to avoid many technical problems. Moreover, the Ikeda-like systems are the natural choice if one wants to keep a resemblance to the classical multilayer perceptron. Therefore, such systems are the first choice when dealing with delay-based ML applications.

However, we do not see any fundamental obstacles to using other types of time-delayed systems. We have made the following amendments in the manuscript in order to emphasize this point:

The type or exact nature of the single neuron is not essential. To facilitate the presentation of the main ideas, we assume that the system state evolves in continuous time according to a differential equation of the general form:

...

Systems of the form (1) are typical for machine learning applications with delay models [13,14,33,20].

- The proposed method seems at the intersection of “conventional” machine learning (use of backpropagation, etc.) and delay-based reservoir computing (single

neuron, etc.). The reader would intuitively assume that the new method intends to leverage from the best of both worlds. A dedicated section where the performance of the Fit-DNN would be discussed in contrast to the two other approaches would be of interest.

Indeed, the Fit-DNN concept incorporates both: delay-based reservoir computing and DNN. In particular, for large node separation θ , the Fit-DNN mimics conventional multilayer perceptrons. Therefore, the performance in terms of accuracy is equivalent in this case. Choosing a smaller θ benefits the overall computation time, but decreases the achievable accuracy. This decrease strongly depends on the considered tasks (see Fig. 4).

Compared to delay-based reservoir computing, our concept radically expands the range of possible applications. We consider this to be one of the main advantages of Fit-DNN. See also our related responses to question 1 of reviewer 1 and question 1 of reviewer 2 above.

To address this issue more strongly in the manuscript, we have made the following changes:

In the caption to Fig. 4, we changed the sentence:

The accuracy obtained in the map limit case $\theta \rightarrow \infty$ is shown by the horizontal black line.

to

The accuracy obtained in the map limit case $\theta \rightarrow \infty$ is shown by the horizontal black line (this corresponds to the classical sparse multilayer perceptron).

We also added a paragraph in the Discussion section:

Another major aspect of the Fit-DNN construction is the importance of the temporal node separation θ . For large node separation θ , the Fit-DNN mimics conventional multilayer perceptrons. Therefore, the performance in terms of accuracy is equivalent in this case. In contrast, choosing a smaller θ benefits the overall computation time, but decreases the achievable accuracy. This decrease strongly depends on the considered tasks (see Fig. 4).

- Could the authors elaborate on the efficiency of their method from the viewpoint of computation time?

Your question has much in common with question 5 of reviewer 1. Please refer to our detailed response to this comment and the description of the corresponding amendments to the manuscript.

~~Deep Learning with a Single Neuron: Folding a Deep Neural Network in Time using Feedback-Modulated Delay Loops~~ Deep Neural Networks using a Single Neuron: Folded-in-Time Architecture using Feedback-Modulated Delay Loops

Florian Stelzer^{1,2,4}, André Röhm³, Raul Vicente⁴, Ingo Fischer³, and Serhiy Yanchuk^{1,*}

¹Institute of Mathematics, Technische Universität Berlin, 10623, Germany

²Department of Mathematics, Humboldt-Universität zu Berlin, 12489, Germany

³Instituto de Física Interdisciplinar y Sistemas Complejos, IFISC (UIB-CSIC), Campus Universitat de les Illes Balears, E-07122 Palma de Mallorca, Spain

⁴Institute of Computer Science, University of Tartu, Tartu, Estonia

*corresponding author

Abstract

Deep neural networks are among the most widely applied machine learning tools showing outstanding performance in a broad range of tasks. We present a method for folding a deep neural network of arbitrary size into a single neuron with multiple time-delayed feedback loops. This single-neuron deep neural network comprises only a single nonlinearity and appropriately adjusted modulations of the feedback signals. The network states emerge in time as a temporal unfolding of the neuron’s dynamics. By adjusting the feedback-modulation within the loops, we adapt the network’s connection weights. These connection weights are determined via a ~~modified~~ back-propagation algorithm ~~that we designed for such types of networks, where both the delay-induced and local network connections must be taken into account~~. Our approach ~~fully recovers~~ can fully represent standard Deep Neural Networks (DNN), encompasses sparse DNNs, and extends the DNN concept toward dynamical systems implementations. The new method, which we call Folded-in-time DNN (Fit-DNN), exhibits promising performance in a set of benchmark tasks.

1 Introduction

Fueled by Deep Neural Networks (DNN), machine learning systems are achieving outstanding results in large-scale problems. The data-driven representations learned by DNNs empower state-of-the-art solutions to a range of tasks in computer vision, reinforcement learning, robotics, health-care, and natural language processing [1, 2, 3, 4, 5, 6, 7, 8, 9]. Their success has also motivated the implementation of DNNs using alternative hardware platforms, such as photonic or electronic

concepts, see, e.g., [10, 11, 12] and references therein. However, so far, these alternative hardware implementations require major technological efforts to realize partial functionalities, and, depending on the hardware platform, the corresponding size of the DNN remains rather limited [12].

Here, we introduce a folding-in-time approach to emulate a full DNN using only a single artificial neuron with feedback-modulated delay loops. Temporal modulation of the signals within the individual delay loops allows realizing adjustable connection weights among the hidden layers. This approach can reduce the required hardware drastically and offers a new perspective on how to construct trainable complex systems: The large network of many interacting elements is replaced by a single element, representing different elements in time by interacting with its own delayed states. We are able to show that our folding-in-time approach is fully equivalent to a feed-forward deep neural network under certain constraints—and that it, in addition, encompasses dynamical systems specific architectures. We name our approach *Folded-in-time Deep Neural Network* or short **Fit-DNN**.

Our approach follows an interdisciplinary mindset that draws its inspiration from the intersection of AI systems, brain-inspired hardware, dynamical systems, and analogue computing. Choosing such a different perspective on DNNs leads to a better understanding of their properties, requirements, and capabilities. In particular, we discuss the nature of our Fit-DNN from a dynamical systems’ perspective. We derive ~~an adapted a~~ back-propagation approach applicable to gradient descent training of Fit-DNNs based on continuous dynamical systems and demonstrate that it provides good performance results in a number of tasks. Our approach will open up new strategies to implement DNNs in alternative hardware.

For the related machine learning method called ‘reservoir computing’ based on fixed recurrent neural networks, folding-in-time concepts have already been successfully developed [13]. Delay-based reservoir computing typically uses a single delay loop configuration and time-multiplexing of the input data to emulate a ring topology. The introduction of this concept led to a better understanding of reservoir computing, its minimal requirements, and suitable parameter conditions. Moreover, it facilitated their implementation on various hardware platforms [13, 14, 15, 16, 17, 18, 19]. In fact, the delay-based reservoir computing concept inspired successful implementations in terms of hardware efficiency [13], processing speed [16, 20, 21], task performance [22, 23], and last, but not least, energy consumption [16, 22].

~~For DNN, a folding-in-time approach has been lacking so far. Our concept of folded-in-time deep neural networks also benefits from time-multiplexing, but uses it in a more intricate manner going conceptually beyond by allowing for the implementation of multi-layer feed-forward neural networks with adaptable hidden layer connections and, in particular, the applicability of the gradient descent method for their training.~~ We present the Fit-DNN concept and show its versatility and applicability ~~to solve real-world by solving benchmark~~ tasks.

2 Results

2.1 A network folded into a single neuron

The traditional Deep Neural Networks consist of multiple layers of neurons coupled in a feed-forward architecture. Implementing their functionality with only a single neuron requires preserving the logical order of the layers while finding a way to sequentialize the operation within the layer. This can only be achieved by temporally spacing out processes that previously acted simultaneously. A single neuron receiving the correct inputs at the correct times sequentially em-

Figure 1: Scheme of the Fit-DNN setup. A nonlinear element (neuron) with a nonlinear function f is depicted by a black circle. The state of the neuron at time t is $x(t)$. The signal $a(t)$ is the sum of the data $J(t)$, bias $b(t)$, and feedback signals. Each feedback loop implements a delay τ_d and a temporal modulation $\mathcal{M}_d(t)$.

ulates each neuron in every layer. The connections that previously linked neighboring layers now instead have to connect the single neuron at different *times*, and thus interlayer links turn into *delay*-connections. The weight of these connections has to be adjustable, and therefore a temporal modulation of these connections is required.

The architecture derived this way is depicted in Fig. 1 and called *Folded-in-time DNN*. The core of the Fit-DNN consists of a single neuron with multiple delayed and modulated feedbacks. The type or exact nature of the single neuron is not essential; ~~we only demand that its~~. To facilitate the presentation of the main ideas, we assume that the system state evolves in continuous time according to a differential equation of the general form:

$$\dot{x}(t) = -\alpha x(t) + f(a(t)), \quad \text{where} \quad (1)$$

$$a(t) = J(t) + b(t) + \sum_{d=1}^D \mathcal{M}_d(t)x(t - \tau_d). \quad (2)$$

Here $x(t)$ denotes the state of the neuron; f is a nonlinear function with the argument $a(t)$ combining the data signal $J(t)$, time-varying bias $b(t)$, and the time-delayed feedback signals $x(t - \tau_d)$ modulated by the functions $\mathcal{M}_d(t)$, see Fig. 1. We explicitly consider multiple loops of different delay lengths τ_d . Due to the feedback loops, the system becomes a so-called *delay dynamical system*, which leads to profound implications for the complexity of its dynamics [24, 25, 26, 27, 28, 29, 30, 31, 32]. Systems of the form (1) are typical for machine learning applications with delay models [13, 14, 33, 20].

Intuitively, the feedback loops in Fig. 1 lead to a reintroduction of information that has already passed through the nonlinearity f . This allows chaining the nonlinearity f many times. While a classical DNN composes its trainable representations by using neurons layer-by-layer, the Fit-DNN achieves the same by reintroducing a feedback signal to the same neuron repeatedly. In each pass, the time-varying bias $b(t)$ and the modulations $\mathcal{M}_d(t)$ on the delay-lines ensure that the time evolution of the system processes information in the desired way. To obtain the data signal $J(t)$ and output \hat{y} we need an appropriate pre- or postprocessing, respectively.

Figure 2: Equivalence of the Fit-DNN using a single neuron with modulated delayed feedbacks to a classical DNN. Panel (a): The neuron state is considered at discrete time points $x_n^\ell := x((\ell - 1)T + n\theta)$. The intervals $((\ell - 1)T, \ell T]$ correspond to layers. Due to delayed feedbacks, non-local connections emerge (color lines). Panel (b) shows a stacked version of the plot in panel (a) with the same active connections. Panel (c) shows the resulting network: it is a rotated version of (b), with additional input and output layers. Black lines indicate connections implied by the temporal ordering of the emulation.

2.2 Equivalence to multi-layer neural networks

To further illustrate how the Fit-DNN is functionally equivalent to a multi-layer neural network, we present Fig. 2 showing the main conceptual steps for transforming the dynamics of a single neuron with multiple delay loops into a DNN. A sketch of the time-evolution of $x(t)$ is presented in Fig. 2a. This evolution is divided into time-intervals of length T , each emulating a hidden layer. In each of the intervals, we choose N points. We use a grid of equidistant timings with small temporal separation θ . For hidden layers with N nodes, it follows that $\theta = T/N$. At each of these temporal grid points $t_n = n\theta$, we treat the system state $x(t_n)$ as an independent variable. Each temporal grid point t_n will represent a node, and $x(t_n)$ its state. We furthermore assume that the data signal $J(t)$, bias $b(t)$, and modulation signals $\mathcal{M}_d(t)$ are step functions with step-lengths θ ; we refer to the Methods Sec. 4 for their precise definitions.

By considering the dynamical evolution of the time-continuous system $x(t)$ only at these discrete temporal grid points t_n (black dots in Fig. 2a), one can prove that the Fit-DNN emulates a classical DNN. To show it formally, we define network nodes x_n^ℓ of the equivalent DNN as

$$x_n^\ell := x((\ell - 1)T + n\theta), \quad (3)$$

with $n = 1, \dots, N$ determining the node's position within the layer, and $\ell = 1, \dots, L$ determining the layer. Analogously, we define the activations a_n^ℓ of the corresponding nodes. Furthermore, we add an additional node $x_{N+1}^\ell := 1$ to take into account the bias. Thus, the points from the original time-intervals T are now described by the vector $x^\ell = (x_1^\ell, \dots, x_N^\ell)$. Figure 2b shows the original time-trace cut into intervals of length T and nodes labeled according to their network position. The representation in Fig. 2c is a rotation of Fig. 2b with the addition of an input and an output layer.

The connections are determined by the dynamical dependencies between the nodes x_n^ℓ . These dependencies can be explicitly calculated either for small or large distance θ . In the case of a large node separation θ , the relations between the network nodes x_n^ℓ is of the familiar DNN shape:

$$x_n^\ell = \alpha^{-1} f(a_n^\ell), \quad (4)$$

$$a^\ell := W^\ell x^{\ell-1}. \quad (5)$$

System (4) is derived in detail in the Supplementary Information. The matrix W^ℓ describes the connections from layer $\ell - 1$ to ℓ and corresponds to the modulated delay-lines in the original single-neuron system. Each of the time-delayed feedback loops leads to a dependence of the state $x(t)$ on $x(t - \tau_d)$, see colored arrows in Fig. 2a. By way of construction, the length of each delay-loop is fixed. Since the order of the nodes (3) is tied to the temporal position, a fixed delay-line cannot connect arbitrary nodes. Rather, each delay-line is equivalent to one diagonal of the coupling matrix W^ℓ . Depending on the number of delay loops D , the network possesses a different connectivity level between the layers. A fully connected Fit-DNN requires $2N - 1$ modulated delay loops, i.e., our connectivity requirement scales *linearly* in the system size N and is entirely independent of L , promising a favorable scaling for hardware implementations.

The time-dependent modulation signals $\mathcal{M}_d(t)$ allow us to set the feedback strengths to zero at certain times. For this work, we limit ourselves to delayed feedback connections, which only link nodes from the neighboring layers, but in principle this limitation could be lifted if more exotic networks were desired. For a visual representation of the connections implied by two sample delay loops, see Fig. 2b and c. The mismatch between the delay τ_d and T determines, which nodes are connected by that particular delay-loop: For $\tau_d < T$ ($\tau_d > T$), the delayed feedback connects a node x_n^ℓ with another node $x_i^{\ell+1}$ in a subsequent layer with $n > i$ ($n < i$), shown with red (yellow) arrows in Fig. 2.

To complete the DNN picture, the activations for the first layer will be rewritten as $a^1 := g(a^{\text{in}}) := g(W^{\text{in}}u)$, where W^{in} is used in the preprocessing of $J(t)$. A final output matrix W^{out} is used to derive the activations of the output layer $a^{\text{out}} := W^{\text{out}}x^L$. We refer to the Methods Sec. 4.2 for a precise mathematical description.

2.3 Dynamical systems perspective: small node separation

For small node separation θ , the Fit-DNN approach goes beyond the standard DNN. Inspired by the method used in [13, 34, 35], we apply the variation of constants formula to solve the linear part of (1) and the Euler discretization for the nonlinear part and obtain the following relations between the nodes up to the first-order terms in θ :

$$x_n^\ell = e^{-\alpha\theta} x_{n-1}^\ell + \alpha^{-1} (1 - e^{-\alpha\theta}) f(a_n^\ell), \quad n = 2, \dots, N, \quad (6)$$

for the layers $\ell = 1, \dots, L$, and nodes $n = 2, \dots, N$. Note, how the first term $e^{-\alpha\theta} x_{n-1}^\ell$ couples each node to the preceding one within the same layer. Furthermore, the first node of each layer ℓ

is connected to the last node of the preceding layer:

$$x_1^\ell = e^{-\alpha\theta} x_N^{\ell-1} + \alpha^{-1}(1 - e^{-\alpha\theta})f(a_1^\ell), \quad (7)$$

where $x_N^0 := x_0 = x(0)$ is the initial state of system (1). Such a dependence reflects the fact that the network was created from a single neuron with time-continuous dynamics. With ~~insufficient-a small~~ node separation θ , each node state residually depends on the preceding one and is not fully independent. These additional ‘inertial’ connections are represented by the black arrows in the network representation in Fig. 2c and are present in the case of small θ .

This second case of small θ may seem like a spurious, superfluous regime that unnecessarily complicates the picture. However, in practice, a small θ directly implies a fast operation—as the time the single neuron needs to emulate a layer is directly given by $N\theta$. We, therefore, expect this regime to be of interest for future hardware implementations. Additionally, while we recover a fully connected DNN using $D = 2N - 1$ delay loops, our simulations show that this is not a strict requirement. Adequate performance can already be obtained with a much smaller number of delay loops. In that case, the Fit-DNN is implementing a particular type of sparse DNNs.

2.4 Back-propagation for Fit-DNN

The Fit-DNN (4) for large θ is the classical multilayer perceptron; hence, the weight gradients can be computed using the classical back-propagation algorithm [36, 37, 3]. ~~However, if~~. If less than the full number of delay-loops is used, the resulting DNN will be sparse. Training sparse DNN is a current topic of research [38, 39], but as a first step, the traditional back-propagation algorithm can still be applied. However, the sparsity does not affect the gradient computation for the weight adaptation.

For a small temporal node separation θ , the Fit-DNN approach differs from the classical multilayer perceptron because it contains additional linear intra-layer connections and additional linear connections from the last node of one hidden layer to the first node of the next hidden layer, see Fig. 2c, black arrows. Nonetheless, the network can be trained by adjusting the input weights W^{in} , the output weights W^{out} , and the non-zero elements of the potentially sparse weight matrices W^ℓ using gradient descent. For this, we employ a ~~modified~~ back-propagation algorithm, described in Sec. 4.3, which takes these additional connections into consideration.

2.5 Benchmark tasks

Since under certain conditions, the Fit-DNN fully recovers a standard DNN (without convolutional layers), the resulting performance will be identical. This is obvious, when considering system (4), since the dynamics are perfectly described by a standard multilayer perceptron. However, the Fit-DNN approach also encompasses the aforementioned cases of short temporal node distance θ and the possibility of using less delay-loops, which translates to a sparse DNN. We report here that the system retains its computational power even in these regimes, i.e., a Fit-DNN can in principle be constructed with few and short delay-loops.

To demonstrate the computational capabilities of the Fit-DNN over these regimes, we considered ~~four-five~~ image classification tasks: MNIST [40], Fashion-MNIST [41], CIFAR-10, CIFAR-100 considering the coarse class labels [42], and the cropped version of SVHN [43]. As a demonstration for a very sparse network, we applied the Fit-DNN to an image denoising task: We added Gaussian noise of intensity $\sigma_{\text{task}} = 1$ to the images of the Fashion-MNIST dataset, which we considered as vectors with values between 0 (white) and 1 (black). Then we clipped the resulting vector entries at the clipping thresholds 0 and 1 in order to obtain noisy grayscale images. The denoising task

Figure 3: Example images for the denoising task. Row (a) contains original images from the Fashion-MNIST data set. Row (b) shows the same images with additional Gaussian noise. These noisy images serve as input data for the trained system. Row (c) shows the obtained reconstructions of the original images.

	(a)	(b)	(c)	(d)
input nodes M	784	3072	3072	784
output nodes P	10	10	20	784
nodes per hidden layer N	100	100	100	100
number of hidden layers L	2	3	3	2
number of delays D	100	100	100	5
node separation θ	0.5	0.5	0.5	0.5
system time scale α	1	1	1	1
initial training rate η_0	0.01	0.0001	0.0001	0.001
training rate scaling factor η_1	10000	1000	1000	500
intensity of training noise σ	0.1	0.01	0.01	–

Table 1: Standard parameters for (a) the MNIST and Fashion-MNIST tasks, (b) the CIFAR-10 and cropped SVHN tasks, ~~and~~ (c) the CIFAR-100 tasks with coarse class labels, and (d) the image denoising task.

is to reconstruct the original images from their noisy versions. Figure 3 shows examples of the original Fashion-MNIST images, their noisy versions, and reconstructed images.

For the tests, we solved the delay system (1) numerically and trained the weights by gradient descent using the ~~modified~~-back-propagation algorithm described in Sec. 4.3. Unless noted otherwise, we operated in the small θ regime, and in general did not use a fully connected network. By nature of the architecture, the choice of delays τ_d is not trivial. We always chose the delays as a multiple of θ , i.e. $\tau_d = n_d \theta$, $d = 1, \dots, D$. The integer n_d can range from 1 to $2N - 1$ and indicates which diagonal of the weight matrix W^ℓ is accessed. After some initial tests, we settled on drawing the numbers n_d from a uniform distribution on the set $\{1, \dots, 2N - 1\}$ without replacement.

If not stated otherwise, we used the activation function $f(a) = \sin(a)$, but the Fit-DNN is in principle agnostic to the type of nonlinearity f that is used. The standard parameters for our numerical tests are listed in Table 1. For further details we refer to the Methods Sec. 4.4.

In Table 2, we show the Fit-DNN performance for different numbers of the nodes $N = 50, 100, 200$, and 400 per hidden layer on the aforementioned tasks. We immediately achieve high success rates on the relatively simple MNIST and Fashion-MNIST tasks. The more challenging CIFAR-10, coarse CIFAR-100 and cropped SVHN tasks obtain lower yet still significant success rates. The confusion matrices (see Supplementary Information) also show that the system tends to confuse similar categories (e.g. ‘automobile’ and ‘truck’). While these results clearly do not rival record state-of-the art performances, they were achieved on a novel and radically different architecture. In particular, the Fit-DNN here only used about half of the available diagonals of the weight matrix and operated in the small θ regime. For the tasks tested, increasing N clearly

N	50	100	200	400	
MNIST	97.31	98.49	98.91	98.97	[%]
Fashion-MNIST	86.61	87.82	88.59	89.18	[%]
CIFAR-10	48.29	51.42	53.94	54.99	[%]
coarse CIFAR-100	29.39	32.73	34.51	35.41	[%]
cropped SVHN	73.45	78.93	80.85	81.38	[%]
denoising	0.0277	0.0254	0.0241	0.0236	[MSE]

Table 2: Fit-DNN performance for classification and denoising tasks; dependence on the number of nodes per hidden layer N . Shown are accuracies [in %] and mean squared error for the denoising task for different N . Increasing N improves the results for all tasks. For the classification tasks with $N = 50$, the number of delays is $D = 99$, for the other cases the standard value $D = 100$ is used. For the denoising task, $D = 5$ is used for all cases.

leads to increased performance. This also serves as a sanity check and proves the scalability of the concept. In particular, note that if implemented in some form of dedicated hardware, increasing the number of nodes per layer N does not increase the number of components needed, solely the time required to run the system. Also note, that the denoising task was solved using only 5 delay-loops. For a network of 400 nodes, this results in an extremely sparse weight matrix W^ℓ . Nonetheless, the system performs well.

Figure 4 shows the performance of the Fit-DNN for the classification tasks and the correctness of the computed gradients for different node separations θ . Since this is one of the key parameters that controls the Fit-DNN, understanding its influences is of vital interest. We also use this opportunity to illustrate the ~~difference between the modified and the classical importance of considering the linear local connections when performing~~ back-propagation ~~algorithm for the gradient descent to compute the weight gradients~~. We applied gradient checking, i.e., the comparison to a numerically computed practically exact gradient, to determine the correctness of the ~~gradient estimates obtained from both back-propagation methods~~ obtained gradient estimates. We also trained the map limit network (4) for comparison, corresponding to a (sparse) multilayer perceptron. In this way, we can also see how the additional intra-layer connections influence the performance for small θ .

The obtained results of Fig. 4 ~~confirm the advantage of the modified~~ show that back-propagation ~~algorithm over the classical one for the Fit-DNN. Specifically, the modified provides good estimates of the gradient over the entire range of θ . They also highlight the strong influence of the local connections. More specifically, taking into account the local connections, the~~ back-propagation algorithm ~~(blue points in Fig. 4)~~ yields correct gradients for large node separations $\theta \geq 4$ and for small node separations $\theta \leq 0.125$ (blue points in Fig. 4). For intermediate node separations, we obtain a rather rough approximation of the gradient, but the cosine similarity between the actual gradient and its approximation is still at least 0.8, i.e., the approximation is good enough to train effectively. In contrast, ~~the classical if local connections are neglected, back-propagation algorithm (red points in Fig. 4) only works for large node separations works only for a large node separation~~ $\theta \geq 4$, where the system approaches the map limit ~~-(red points in Fig. 4)~~. Consequently, we obtain competitive accuracies for the MNIST and the Fashion-MNIST tasks even for small θ if we use ~~the modified~~ back-propagation ~~method with properly included local connections~~. When we apply the Fit-DNN to the more challenging CIFAR-10, coarse CIFAR-100 and cropped SVHN tasks, small node separations affect the accuracies negatively. However, we still ~~observe a significant advantage of the modified over the classical algorithm~~ obtain reasonable results for moderate node separations.

Further numerical results regarding the number of hidden layers L , the number of delays D ,

Figure 4: Fit-DNN performance for classification and denoising tasks; dependence on the node separation θ . Shown are accuracies of the classification tasks by employing the modified back-propagation algorithm taking the local coupling into consideration (blue points), and classical neglecting them (red points) back-propagation algorithm; panels (a, c, e, and g). The accuracy obtained in the map limit case $\theta \rightarrow \infty$ is shown by the horizontal black line (this corresponds to the classical sparse multilayer perceptron). Lower panels show the cosine similarities between the numerically computed approximation of the exact gradient and the gradient obtained from the modified by back-propagation with (blue points) and classical or without (red) back-propagation method local connections.

and the role of the activation function f are presented in detail in the Supplementary Information. We find that the optimal choice of L depends on the node separation θ . Our findings suggest that for small θ , one should choose a smaller number of hidden layers than for the map limit case $\theta \rightarrow \infty$. The effect of the number of delays D depends on the task. We found that a small number of delays is sufficient for the denoising task: the mean squared error remains constant when varying D between 5 and 40. For the CIFAR-10 task, a larger number of delays is necessary to obtain optimal results. If we use the standard parameters from Table 1, we obtain the highest CIFAR-10 accuracy for $D = 125$ or larger. This could likely be explained by the different requirements of these tasks: While the main challenge for denoising is to filter out unwanted points, the CIFAR-10 task requires attention to detail. Thus, a higher number of delay-loops potentially helps the system to learn a more precise representation of the target classes. By comparing the Fit-DNN performance for different activation functions, we also confirmed that the system performs similarly well for the sine $f(a) = \sin(a)$, the hyperbolic tangent $f(a) = \tanh(a)$, and the ReLU function $f(a) = \max\{0, a\}$.

3 Discussion

3.1 General aspects of the folding-in-time concept

We have designed a method for complete ~~Folding-in-time of a multi-layer~~ folding-in-time of a multilayer feed-forward DNN. This Fit-DNN approach requires only a single neuron with feedback-modulated ~~delay-loops~~ delay loops. Via a temporal sequentialization of the nonlinear operations, an arbitrarily deep or wide DNN can be realized. We also naturally arrive at ~~slight modifications, such as sparse DNN or DNNs, which contain such~~ modifications as sparse DNNs or DNNs with additional inertial connections. ~~For this, we have developed a modified back-propagation method for~~ We have demonstrated that gradient descent training of the coupling weights is not significantly interfered by these additional local connections.

~~Even for the case of~~ Extending machine-learning architectures to be compatible with a dynamical delay-system perspective can help fertilize both fundamental research and applications. For example, the idea of time-multiplexing a recurrent network into a single element was introduced in [13] and had a profound effect on understanding and boosting the reservoir computing concept. In contrast to the time-multiplexing of a fixed recurrent network for reservoir computing, here we use the extended folding-in-time technique to realise feed-forward DNNs, thus implementing layers with adaptive connection weights. Compared to delay-based reservoir computing, our concept focuses on the different and extended range of possible applications of DNNs.

3.2 Dynamical systems perspective

From a general perspective, our approach provides an alternative view on neural networks: the entire topological complexity of the feed-forward multilayer neural networks can be folded into the temporal domain by the delay-loop architecture. This exploits the prominent advantage of time-delay systems that ‘space’ and ‘time’ can intermingle, and delay systems are known to have rich spatio-temporal properties [44, 32, 45, 46]. This work significantly extends this spatio-temporal equivalence and its application while allowing the evaluation of neural networks with the tools of delay systems analysis [26, 30, 47, 48]. In particular, we show how the transition from the time-continuous view of the physical system, i.e. the delay-differential equation, to the time-discrete feed-forward DNN can be made.

Our concept also differs clearly from the construction of neural networks from ordinary differential equations [49, 50, 51]. Its main advantage is that delay systems inherently possess an infinite-dimensional phase space. As a result, just one neuron with feedback is sufficient to fold the entire complexity of the network.

3.3 Sparsity, scaling and node separation

It has been shown that dynamic sparsity [38, 39] can outperform dense networks and, fundamentally, Fit-DNNs are intrinsically compatible with certain kinds of sparsity. However, in our approach, removing or adding a delay loop would change an entire diagonal in the hidden weight matrices. Therefore, sparsity training algorithms such as [38, 39] and related works are not directly applicable to the Fit-DNN. Our preliminary tests have shown that removing the weights of a diagonal at the same time disturbs the previous training too much, so the method fails. Nevertheless, we expect that it is possible to find a suitable method to optimize the choice of delays. Therefore, further investigation of specific sparsity training methods for the Fit-DNN would be very welcome. One

candidate for such a method could be pruning by slowly fading diagonals that contain weaker connections on average.

Even with a fixed sparse connectivity, we can perform image classification using only a single dynamical neuron. This case, in particular, highlights one of the most exciting aspects of the Fit-DNN architecture: Many hardware implementations of DNNs or related systems have suffered from the large amount of elements that need to be implemented: the active neurons as well as the connections with adjustable weights. The Fit-DNN overcomes both of these limitations: no matter how many neurons are functionally desired, physically we only require a single one. Even though we advocate for sparse connectivity in this paper, a fully connected DNN would require only linearly scaling only require a linear scaling of the number of delay loops with the number of nodes per layer N . This is in strong contrast represents a major advantage as compared to directly implemented networks, where the number of connections grows quadratically. Where Thus, where it is acceptable to use sparse networks, increasing the number of layers L or the number of nodes per layer N for the Fit-DNN only requires more time, but not more elements. At the same time, since only one nonlinear node and, in principle, one fast read-out element are absolutely required in our approach, ultrafast components might be used that would be unrealistic to use in full DNN implementations. hardware elements.

Another major aspect of the Fit-DNN construction is the importance of the temporal node separation θ . For large node separation θ , the Fit-DNN mimics conventional multilayer perceptrons. Therefore, the performance in terms of accuracy is equivalent in this case. In contrast, choosing a smaller θ benefits the overall computation time, but decreases the achievable accuracy. This decrease strongly depends on the considered tasks (see Fig. 4).

3.4 Potential for hardware implementation

In addition to providing a dynamical systems perspective on DNNs, Fit-DNNs can also serve as blueprints for specialized DNN hardware. The Fit-DNN approach is agnostic concerning the type of nonlinearity, enabling flexibility of implementations. A suitable candidate for such a hardware implementation could be a photonic neuromorphic implementation [13, 14, 15, 16, 52, 53, 20], where a fast artificial neuron can be realized with the Gigahertz timescale range. Photonic systems have already been used to construct delay-based reservoir computers, which is where the original idea of folding a neural network in time was first introduced [13]. The possibility of folding a recurrent network into a single element already had a profound effect on understanding these systems. In contrast to the folding-in-time of a recurrent network for reservoir computing, here we use the technique in a different way to realize a (multi-layer) feed-forward network architecture with adaptive connection weights. While more effort is required, the resulting Fit-DNN scheme can successfully represent a DNN and potentially go beyond those. Moreover, in In retrospect, it is quite clear how instrumental the reduced hardware requirement of a delay-based approach was in stimulating the current ecosystem of reservoir computing implementations. For example, the delay-based reservoir computing has been successfully implemented using electronic systems, magnetic spin systems, MEMS, acoustic, and other platforms. We hope that for the much larger community around DNNs, a similar similarly stimulating effect can be achieved with the Fit-DNN approach we presented here, since it also drastically reduces the cost and complexity for hardware-based DNNs.

From a general perspective, our approach provides an alternative view on neural networks: the whole topological complexity of the feed-forward multilayer neural networks can be folded into the temporal domain by the delay-loop architecture. This uses the prominent advantage of time-delay systems that ‘space’ and ‘time’ can intermingle, and delay systems are known to possess

rich spatio-temporal features [44, 32, 45, 46]. This work extends this spatio-temporal equivalence and its application significantly. Moreover, such an equivalence allows for the evaluation of neural networks with the tools of delay systems analysis [26, 30, 47, 48]

Certainly, realizations on different hardware platforms face different challenges. In the following, we exemplify the requirements for a photonic (optoelectronic) scheme. Such an implementation requires only one light source, a few fiber couplers, and optical fibers of different lengths. The modulations of the delay loops can be implemented using Mach-Zehnder intensity modulators. Finally, only two fast photodetectors (one for all delay loops and one for the output) would be required, as well as an optical amplifier or an electrical amplifier which could be used to compensate for roundtrip losses. Those are all standard telecommunication components. The conversion from optical to electrical signals can be done extremely fast, faster than the clock rate of today's fast electronic processors, and only two photodetectors are needed, regardless of the number of virtual nodes and number of delay loops.

3.5 Trade-Offs

Since only one nonlinear node and one fast read-out element are absolutely necessary in our approach, ultrafast components could be used that would be unrealistic or too expensive for full DNN implementations. At the same time, since the single nonlinear element performs all nonlinear operations sequentially with node separation θ , parallelization cannot be applied in this approach. The overall processing time scales linearly with the total number of nodes LN and with the node separation θ . Possible ways to address this property that could represent a limitation in certain applications include the use of a small node separation θ [13] or multiple parallel copies of Fit-DNNs. In this way, a tradeoff between the number of required hardware components and the amount of parallel processing is possible. At the same time, the use of a single nonlinear node comes with the advantage of almost perfect homogeneity of all folded nodes, since they are realised by the same element.

Our concept also differs clearly from the construction of neural networks from ordinary differential equations [49, 50, 51]. Our concept's main advantage is that delay systems inherently possess an infinite-dimensional phase space. As a result, just one neuron is sufficient to fold the whole complexity of the network architecture. We would also like to point out that the potential use of very fast hardware components is accompanied by a possibility of fast inference. However, a fast hardware implementation of the Fit-DNN will not accelerate the training process, because a traditional computer is still required, at least for the back-propagation of errors. If the forward propagation part of the training process is also performed on a traditional computer, the delay equation must be solved numerically for each training step, leading to a significant increase in training time. Therefore, the presented method is most suitable when fast inference and/or high hardware efficiency are prioritized. We would like to point out that the integration of the training process into the hardware-part could be addressed in future extensions of our concept.

3.6 Outlook

We have presented a minimal and concise model, but already a multitude of potential extensions are apparent for future studies. For instance, one can implement different layer sizes, multiple nonlinear elements, and ~~several other structures~~. combine different structures such as recurrent neural networks with trainable hidden layers.

Incorporating additional neurons (spatial nodes) might even enable finding the optimal trade-off between spatial and temporal nodes, depending on the chosen platform and task. Also, we

envison building a hierarchical neural network consisting of interacting neurons, each of them folding a separate Fit-DNN in the temporal domain. Altogether, starting with the design used in this work, we might unlock a plethora of neural network architectures.

Finally, our approach encourages further cross-fertilization among different communities. While the spatio-temporal equivalence and the peculiar properties of delay-systems may be known in the dynamical systems community, so far, no application to DNNs had been considered. Conversely, the Machine Learning core idea is remarkably powerful, but usually not formulated to be compatible with continuous-time delay-dynamical systems. The Fit-DNN approach unifies these perspectives—and in doing so, provides a concept that is promising for those seeking a different angle to obtain a better understanding or to implement the functionality of DNNs in dedicated hardware.

4 Methods

4.1 The delay system and the signal $a(t)$

The delay system (1) is driven by a signal $a(t)$ which is defined by Eq. (2) as a sum of a data signal $J(t)$, modulated delayed feedbacks $\mathcal{M}_d(t)x(t - \tau_d)$, and a bias $b(t)$. In the following, we describe the components in detail.

(i) The input signal. Given an input vector $(u_1, \dots, u_M)^T \in \mathbb{R}^M$, a matrix $W^{\text{in}} \in \mathbb{R}^{N \times (M+1)}$ of input weights w_{nm}^{in} and an input scaling function g , we define

$$J(t) := g \left(w_{n,M+1}^{\text{in}} + \sum_{m=1}^M w_{nm}^{\text{in}} u_m \right), \quad (8)$$

for $(n-1)\theta < t \leq n\theta$ and $n = 1, \dots, N$. This rule defines the input signal $J(t)$ on the time interval $(0, T]$, whereas $J(t) = 0$ for the other values of t . Such a restriction ensures that the input layer connects only to the first hidden layer of the Fit-DNN. Moreover, $J(t)$ is a step function with the step lengths θ .

(ii) The feedback signals. System (1) contains D delayed feedback terms $\mathcal{M}_d(t)x(t - \tau_d)$ with the delay times $\tau_1 < \dots < \tau_D$, which are integer multiples of the stepsize $\tau_d = n_d\theta$, $n_d \in \{1, \dots, 2N - 1\}$.

The modulation functions \mathcal{M}_d are defined interval-wise on the layer intervals $((\ell - 1)T, \ell T]$. In particular, $\mathcal{M}_d(t) := 0$ for $t \leq T$. For $(\ell - 1)T + (n - 1)\theta < t \leq (\ell - 1)T + n\theta$ with $\ell = 2, \dots, L$ and $n = 1, \dots, N$, we set

$$\mathcal{M}_d(t) := v_{d,n}^\ell. \quad (9)$$

Thus, the modulation functions $\mathcal{M}_d(t)$ are step functions with step length θ . The numbers $v_{d,n}^\ell$ play the role of the connection weights from layer $\ell - 1$ to layer ℓ . More precisely, $v_{d,n}^\ell$ is the weight of the connection from the $(n + N - n_d)$ -th node of layer $\ell - 1$ to the n -th node of layer ℓ . Section 4.2 below explains how the modulation functions translate to the hidden weight matrices W^ℓ . In order to ensure that the delay terms connect only consecutive layers, we set $v_{d,n}^\ell = 0$ whenever $n_d < n$ or $n_d > n + N - 1$ holds.

(iii) **The bias signal.** Finally, the bias signal $b(t)$ is defined as the step function

$$b(t) := b_n^\ell, \quad \text{for } (\ell - 1)T + (n - 1)\theta < t \leq (\ell - 1)T + n\theta, \quad (10)$$

where $n = 1, \dots, N$ and $\ell = 2, \dots, L$. For $0 \leq t \leq T$, we set $b(t) := 0$ because the bias weights for the first hidden layer are already included in W^{in} , and thus in $J(t)$.

4.2 Network representation for small node separation θ

In this section, we provide details to the network representation of the Fit-DNN which was outlined in Sec. 2. The delay system (1) is considered on the time interval $[0, LT]$. As we have shown in Sec. 2, it can be considered as multi-layer neural network with L hidden layers, represented by the solution on sub-intervals of length T . Each of the hidden layers consists of N nodes. Moreover, the network possesses an input layer with M nodes and an output layer with P nodes. The input and hidden layers are derived from the system (1) by a discretization of the delay system with step length θ . The output layer is obtained by a suitable readout function on the last hidden layer.

We first construct matrices $W^\ell = (w_{nj}^\ell) \in \mathbb{R}^{N \times (N+1)}$, $\ell = 2, \dots, L$, containing the connection weights from layer $\ell - 1$ to layer ℓ . These matrices are set up as follows: Let $n'_d := n_d - N$, then $w_{n, n-n'_d}^\ell := v_{d,n}^\ell$ define the elements of the matrices W^ℓ . All other matrix entries (except the last column) are defined to be zero. The last column is filled with the bias weights $b_1^\ell, \dots, b_N^\ell$. More specifically,

$$w_{nj}^\ell := \delta_{N+1,j} b_n^\ell + \sum_{d=1}^D \delta_{n-n'_d,j} v_{d,n}^\ell, \quad (11)$$

where $\delta_{n,j} = 1$ for $n = j$, and zero otherwise. The structure of the matrix W^ℓ is illustrated in the Supplementary Information.

Applying the variation of constants formula to system (1) yields for $0 \leq t_0 < t \leq TL$:

$$x(t) = e^{-\alpha(t-t_0)} x(t_0) + \int_{t_0}^t e^{\alpha(s-t)} f(a(s)) \underline{d} s. \quad (12)$$

In particular, for $t_0 = (\ell - 1)T + (n - 1)\theta$ and $t = (\ell - 1)T + n\theta$ we obtain

$$x_n^\ell = e^{-\alpha\theta} x_{n-1}^\ell + \int_{t_0}^{t_0+\theta} e^{\alpha(s-(t_0+\theta))} f(a(s)) \underline{d} s, \quad (13)$$

where $a(s)$ is given by (2). Note that the functions $\mathcal{M}_d(t)$, $b(t)$, and $J(t)$ are step functions which are constant on the integration interval. Approximating $x(s - \tau_d)$ by the value on the right θ -grid point $x(t - \tau_d) \approx x((\ell - 1)T + n\theta - n_d\theta)$ directly yields the network equation (6).

4.3 Application to machine learning and a **modified** back-propagation algorithm

We apply the system to two different types of machine learning tasks: image classification and image denoising. For the classification tasks, the size P of the output layer equals the number of classes. We choose f^{out} to be the softmax function, i.e.

$$\hat{y}_p = f_p^{\text{out}}(a^{\text{out}}) = \frac{\exp(a_p^{\text{out}})}{\sum_{q=1}^P \exp(a_q^{\text{out}})}, \quad p = 1, \dots, P. \quad (14)$$

If the task is to denoise a greyscale image, the number of output nodes P is the number of pixels of the image. In this case, clipping at the bounds 0 and 1 is a proper choice for f^{out} , i.e.

$$\hat{y}_p = f_p^{\text{out}}(a^{\text{out}}) = \begin{cases} 0, & \text{if } a_p^{\text{out}} < 0, \\ a_p^{\text{out}}, & \text{if } 0 \leq a_p^{\text{out}} \leq 1, \\ 1, & \text{if } a_p^{\text{out}} > 1. \end{cases} \quad (15)$$

‘Training the system’ means finding a set of training parameters, denoted by the vector \mathcal{W} , which minimizes a given loss function $\mathcal{E}(\mathcal{W})$. Our training parameter vector \mathcal{W} contains the input weights w_{nm}^{in} , the non-zero hidden weights w_{nj}^{ℓ} , and the output weights w_{pn}^{out} . The loss function must be compatible with the problem type and with the output activation. For the classification task, we use the cross-entropy loss function

$$\mathcal{E}_{\text{CE}}(\mathcal{W}) := - \sum_{k=1}^K \sum_{p=1}^P y_p(k) \ln(\hat{y}_p(k)) = - \sum_{k=1}^K \ln(\hat{y}_{p_t(k)}(k)), \quad (16)$$

where K is the number of examples used to calculate the loss and $p_t(k)$ is the target class of example k . For the denoising tasks, we use the rescaled mean squared error (MSE)

$$\mathcal{E}_{\text{MSE}}(\mathcal{W}) := \frac{1}{2K} \sum_{k=1}^K \sum_{p=1}^P (\hat{y}_p(k) - y_p(k))^2. \quad (17)$$

We train the system by stochastic gradient descent, i.e. for a sequence of training examples $(u(k), y(k))$ we modify the training parameter iteratively by the rule

$$\mathcal{W}_{k+1} = \mathcal{W}_k - \eta(k) \nabla \mathcal{E}(\mathcal{W}_k, u_k, y_k), \quad (18)$$

where $\eta(k) := \min(\eta_0, \eta_1/k)$ is a decreasing training rate.

If the node separation θ is sufficiently large, the local connections within the network become insignificant, and the gradient $\nabla \mathcal{E}(\mathcal{W})$ can be calculated using the classical back-propagation algorithm for multilayer perceptrons. Our numerical studies show that this works well if $\theta \geq 8$ $\theta > 4$ for the considered examples. For smaller node separations, we need to ~~employ a modified back-propagation algorithm~~ take the emerging local connections into account. In the following, we ~~briefly describe and compare both methods~~ first describe the classical algorithm, which can be used in the case of large \$\theta\$. Then we formulate the the back-propagation algorithm for the Fit-DNN with significant local node couplings.

The classical back-propagation algorithm ~~is~~ can be derived by considering a multilayer neural network as a composition of functions

$$\hat{y} = f^{\text{out}}(a^{\text{out}}(a^L(\dots(a^1(a^{\text{in}}(u)))))) \quad (19)$$

and applying the chain rule. The first part of the algorithm is to iteratively compute partial derivatives of the loss function \mathcal{E} w.r.t. the node activations, the so called error signals, for the output layer

$$\delta_p^{\text{out}} := \frac{\partial \mathcal{E}(a^{\text{out}})}{\partial a_p^{\text{out}}} = \frac{\partial \mathcal{E}(a^{\text{out}})}{\partial a_p^{\text{out}}} \quad (20)$$

$$\underline{\delta}_n^L := \frac{\partial \mathcal{E}(a^L)}{\partial a_n^L} = f'(a_n^L) \sum_{p=1}^P \delta_p^{\text{out}} w_{pn}^{\text{out}}, \underline{\delta}_n^\ell := \frac{\partial \mathcal{E}(a^\ell)}{\partial a_n^\ell} = f'(a_n^\ell) \sum_{i=1}^N \delta_i^{\ell+1} w_{in}^\ell, \ell = L-1, \dots, 1. \quad (21)$$

for $p = 1, \dots, P$, and for the hidden layers

$$\underline{\underline{\delta_n^L := \frac{\partial \mathcal{E}(a^L)}{\partial a_n^L} = f'(a_n^L) \sum_{p=1}^P \delta_p^{\text{out}} w_{pn}^{\text{out}},}} \quad (22)$$

$$\underline{\underline{\delta_n^\ell := \frac{\partial \mathcal{E}(a^\ell)}{\partial a_n^\ell} = f'(a_n^\ell) \sum_{i=1}^N \delta_i^{\ell+1} w_{in}^\ell, \quad \ell = L-1, \dots, 1.}} \quad (23)$$

for $n = 1, \dots, N$. Then, the partial derivatives of the loss function w.r.t. the training parameters can be calculated:

$$\underline{\underline{\frac{\partial \mathcal{E}}{\partial w_{nm}^{\text{in}}} = \delta_n^1 u_m g'(a_n^{\text{in}}), \frac{\partial \mathcal{E}}{\partial w_{nj}^\ell} = \delta_n^\ell x_j^{\ell-1}, \ell = 2, \dots, L, \frac{\partial \mathcal{E}}{\partial w_{pn}^{\text{out}}} \frac{\partial \mathcal{E}(\mathcal{W})}{\partial w_{pn}^{\text{out}}} = \delta_p^{\text{out}} x_n^L,}} \quad (24)$$

for $n = 1, \dots, N+1$ and $p = 1, \dots, P$,

$$\underline{\underline{\frac{\partial \mathcal{E}(\mathcal{W})}{\partial w_{nj}^\ell} = \delta_n^\ell x_j^{\ell-1},}} \quad (25)$$

for $\ell = 2, \dots, L$, $j = 1, \dots, N+1$ and $n = 1, \dots, N$, and

$$\underline{\underline{\frac{\partial \mathcal{E}(\mathcal{W})}{\partial w_{nm}^{\text{in}}} = \delta_n^1 g'(a_n^{\text{in}}) u_m,}} \quad (26)$$

for $m = 1, \dots, M+1$ and $n = 1, \dots, N$. For details, see [54] or [3].

Taking into account the additional linear connections ~~into account~~, we need to change the way we calculate the error signals δ_n^ℓ for the hidden layers. Strictly speaking, we cannot consider ~~a^ℓ the loss \mathcal{E} as a function of $a^{\ell-1}$. Hence, we employ a modified approach to derive the back-propagation algorithm for this case. Our approach is to define extended activation vectors $\bar{a}^\ell \in \mathbb{R}^{N+1}$ with $\bar{a}_n^\ell := a_n^\ell$ for $n = 1, \dots, N$ and $\bar{a}_{N+1}^\ell := x_N^{\ell-1}$ for $\ell > 1$ and $\bar{a}_{N+1}^1 := x_0$. The extended activation vector \bar{a}^ℓ can be considered as a function of $\bar{a}^{\ell-1}$, thus the activation vector a^ℓ , for $\ell = 1, \dots, L$, because there are connections skipping these vectors. Also, Eq. (19) becomes invalid. Moreover, nodes of the same layer are connected to each other. However, the network has still a pure feed-forward structure, and hence, we can apply the chain rule to the composition of functions~~

$$\underline{\underline{\hat{y} = f^{\text{out}}(a^{\text{out}}(\bar{a}^L(\dots(\bar{a}^1(a^{\text{in}}(u))))))}.}} \quad (27)$$

back-propagation to calculate the error signals node by node. We obtain the following algorithm ~~to compute the gradient~~.

Step 1: Compute

$$\underline{\underline{\Delta \delta_p^{\text{out}} := \frac{\partial \mathcal{E}(a^{\text{out}})}{\partial a_p^{\text{out}}} = \delta_p^{\text{out}} \frac{\partial \mathcal{E}}{\partial a_p^{\text{out}}} = \hat{y}_p - y_p,}} \quad (27)$$

for $p = 1, \dots, P$.

Step 2: Let $\Phi := \alpha^{-1}(1 - e^{-\alpha\theta})$. Compute

$$\Delta_n^L := \partial_n \mathcal{E}(\bar{a}^L) = \Phi f'(a_n^L) \sum_{p=1}^P \Delta_p^{\text{out}} \sum_{j=n}^N w_{pj}^{\text{out}} e^{-\alpha\theta(j-n)},$$

for $n = 1, \dots, N$, and

$$\Delta_{N+1}^L := \partial_{N+1} \mathcal{E}(\bar{a}^L) = \sum_{p=1}^P \Delta_p^{\text{out}} \sum_{j=1}^N w_{pj}^{\text{out}} e^{-\alpha\theta j}.$$

For $\ell = L-1, \dots, 1$, compute

$$\Delta_n^\ell := \partial_n \mathcal{E}(\bar{a}^\ell) = \Phi f'(a_n^\ell) \left[\Delta_{N+1}^{\ell+1} e^{-\alpha\theta(N-n)} + \sum_{i=1}^N \Delta_i^{\ell+1} \sum_{j=n}^N w_{ij}^{\ell+1} e^{-\alpha\theta(j-n)} \right],$$

for $n = 1, \dots, N$, and

$$\Delta_{N+1}^\ell := \partial_{N+1} \mathcal{E}(\bar{a}^\ell) = \Delta_{N+1}^{\ell+1} e^{-\alpha\theta N} + \sum_{i=1}^N \Delta_i^{\ell+1} \sum_{j=1}^N w_{ij}^{\ell+1} e^{-\alpha\theta j}.$$

Compute the partial the error derivatives w.r.t. the output weights node states of the last hidden layer

$$\frac{\partial \mathcal{E}}{\partial w_{pn}^{\text{out}}} = \Delta_N^L := \frac{\partial \mathcal{E}}{\partial x_N^L} = \sum_{p=1}^P \delta_p^{\text{out}} \underline{x_n^L} \underline{w_{pN}^{\text{out}}}, \quad (28)$$

and

$$\Delta_n^L := \frac{\partial \mathcal{E}}{\partial x_n^L} = \Delta_{n+1}^L e^{-\alpha\theta} + \sum_{p=1}^P \delta_p^{\text{out}} w_{pn}^{\text{out}}, \quad (29)$$

for $n = 1, \dots, N+1$ and $p = 1, \dots, P$. Compute the partial $n = N-1, \dots, 1$. Then compute the error derivatives w.r.t. the hidden weights node activations

$$\frac{\partial \mathcal{E}}{\partial w_{nj}^\ell} \delta_n^L := \frac{\partial \mathcal{E}}{\partial a_n^L} = \Delta_n^{\ell L} \Phi f'(a_n^L) \underline{x_j^{\ell-1}}, \quad (30)$$

for $j = 1, \dots, N+1$, $n = 1, \dots, N$ and $\ell = 2, \dots, L$. $n = 1, \dots, N$. Compute the partial

Step 3: Repeat the same calculations as in step 2 iteratively for the remaining hidden layers $\ell = L - 1, \dots, 1$, while keeping the connection between the nodes x_N^ℓ and $x_1^{\ell+1}$ in mind. That is, compute

$$\Delta_N^\ell := \frac{\partial \mathcal{E}}{\partial x_N^\ell} = \Delta_1^{\ell+1} e^{-\alpha\theta} + \sum_{i=1}^N \delta_i^{\ell+1} w_{iN}^{\ell+1}, \quad (31)$$

and

$$\Delta_n^\ell := \frac{\partial \mathcal{E}}{\partial x_n^\ell} = \Delta_{n+1}^\ell e^{-\alpha\theta} + \sum_{i=1}^N \delta_i^{\ell+1} w_{in}^{\ell+1}, \quad (32)$$

for $n = N - 1, \dots, 1$. Computing the error derivatives w.r.t. the **input weights node activations** works exactly as for the last hidden layer:

$$\frac{\partial \mathcal{E}}{\partial w_{nm}^{\text{in}}} \delta_n^\ell := \frac{\partial \mathcal{E}}{\partial a_n^\ell} = \Delta_n^1 g_n^\ell \Phi f'(a_{-n}^{\text{in}}) u_m, \quad (33)$$

for $m = 1, \dots, M + 1$ and $n = 1, \dots, N$.

Step 4: Calculate weight gradient using Eqs. (24)–(26).

~~For a detailed derivation of this algorithm~~ The above formulas can be derived by the chain rule. Note that many of the weights contained in the sums in Eq. (31) and Eq. (32) are zero when the weight matrices for the hidden layers are sparse. In this case, one can exploit the fact that the non-zero weights are arranged on diagonals and rewrite the sums accordingly to accelerate the computation:

$$\sum_{i=1}^N \delta_i^{\ell+1} w_{in}^{\ell+1} = \sum_{\substack{d=1 \\ 1 \leq n+n'_d \leq N}}^D \delta_{n+n'_d}^{\ell+1} v_{d,n+n'_d}^{\ell+1} \quad (34)$$

For details we refer to the Supplementary Information.

4.4 Data augmentation, input processing and initialization

For all classification tasks, we performed an augmentation of the training input data by adding a small Gaussian noise to the images and by pixel jittering, i.e., randomly shifting the images by at most one pixel horizontally, vertically, or diagonally. For the CIFAR-10 ~~task~~/100 tasks, we also applied a random rotation of maximal $\pm 15^\circ$ and a random horizontal flip with the probability 0.5 to the training input images. Further, we used dropout [55] with a dropout rate of 1% for the CIFAR-10 ~~task~~/100 tasks. For the denoising task, we performed no data augmentation.

Moreover, for the ~~four~~ five classification tasks, we used the input preprocessing function $g(a) = \tanh(a)$. For the denoising task, we applied no nonlinear input preprocessing, i.e. $g(a) = a$. The weights were always initialized by Xavier initialization [56]. In all cases, we used 100 training epochs.

Data availability

In this paper we built on four five publicly available datasets: the MNIST dataset [40], the Fashion-MNIST dataset [41], the CIFAR-10 dataset/100 datasets [42], and the cropped version of the SVHN dataset [43]. All datasets are public and openly accessible online at <http://yann.lecun.com/exdb/mnist/>, <https://github.com/zalandoresearch/fashion-mnist>, <https://www.cs.toronto.edu/~kriz/cifar.html>, <http://ufldl.stanford.edu/housenumbers/>.

Code availability

The source code to reproduce the results of this study is freely available on GitHub: <https://github.com/flori-stelzer/deep-learning-delay-system>.

Acknowledgements

F.S. and S.Y. acknowledge funding by the "Deutsche Forschungsgemeinschaft" (DFG) in the framework of the project 411803875 and IRTG 1740. A.R. and I.F. acknowledge the Spanish State Research Agency, through the María de Maeztu Program for Units of Excellence in R & D (No. MDM-2017-0711). R.V. thanks the financial support from the Estonian Centre of Excellence in IT (EXCITE) funded by the European Regional Development Fund, through the research grant TK148.

Author Contributions

All authors contributed extensively to the work presented in this paper and to the writing of the manuscript.

Competing Interests statement

The authors declare no competing interests.

References

- [1] Lecun, Y., Bengio, Y. & Hinton, G. Deep learning. *Nature* **521**, 436–444 (2015).
- [2] Schmidhuber, J. Deep learning in neural networks: An overview. *Neural Networks* **61**, 85–117 (2015).
- [3] Goodfellow, I., Bengio, Y. & Courville, A. *Deep Learning* (MIT Press, Cambridge, Massachusetts, London, England, 2016).
- [4] Esteva, A. *et al.* Dermatologist-level classification of skin cancer with deep neural networks. *Nature* **542**, 115–118 (2017).
- [5] Jaderberg, M. *et al.* Human-level performance in 3d multiplayer games with population-based reinforcement learning. *Science* **364**, 859–865 (2019).

- [6] Neftci, E. O. & Averbeck, B. B. Reinforcement learning in artificial and biological systems. *Nature Machine Intelligence* **1**, 133–143 (2019).
- [7] Bonardi, A., James, S. & Davison, A. J. Learning one-shot imitation from humans without humans. *IEEE Robotics and Automation Letters* **5**, 3533–3539 (2020).
- [8] Wei, G. Protein structure prediction beyond alphafold. *Nature Machine Intelligence* **1**, 336–337 (2019).
- [9] Brown, T. B. *et al.* Language models are few-shot learners. Preprint at <https://arxiv.org/abs/2005.14165> (2020).
- [10] Misra, J. & Saha, I. Artificial neural networks in hardware: A survey of two decades of progress. *Neurocomputing* **74**, 239 – 255 (2010).
- [11] Schuman, C. D. *et al.* A survey of neuromorphic computing and neural networks in hardware. Preprint at <https://arxiv.org/abs/1705.06963> (2017).
- [12] De Marinis, L., Cococcioni, M., Castoldi, P. & Andriolli, N. Photonic neural networks: A survey. *IEEE Access* **7**, 175827–175841 (2019).
- [13] Appeltant, L. *et al.* Information processing using a single dynamical node as complex system. *Nat. Commun.* **2**, 468 (2011).
- [14] Larger, L. *et al.* Photonic information processing beyond Turing: an optoelectronic implementation of reservoir computing. *Optics Express* **20** (2012).
- [15] Duport, F., Schneider, B., Smerieri, A., Haelterman, M. & Massar, S. All-optical reservoir computing. *Optics Express* **20**, 22783–22795 (2012).
- [16] Brunner, D., Soriano, M. C., Mirasso, C. R. & Fischer, I. Parallel photonic information processing at gigabyte per second data rates using transient states. *Nature Communications* **4**, 1364 (2013).
- [17] Torrejon, J. *et al.* Neuromorphic computing with nanoscale spintronic oscillators. *Nature* **547**, 428–431 (2017).
- [18] Haynes, N. D., Soriano, M. C., Rosin, D. P., Fischer, I. & Gauthier, D. J. Reservoir computing with a single time-delay autonomous boolean node. *Phys. Rev. E* **91**, 020801 (2015).
- [19] Dion, G., Mejaouri, S. & Sylvestre, J. Reservoir computing with a single delay-coupled nonlinear mechanical oscillator. *Journal of Applied Physics* **124**, 152132 (2018).
- [20] Larger, L. *et al.* High-speed photonic reservoir computing using a time-delay-based architecture: Million words per second classification. *Physical Review X* **7**, 1–14 (2017).
- [21] Bueno, J., Brunner, D., Soriano, M. C. & Fischer, I. Conditions for reservoir computing performance using semiconductor lasers with delayed optical feedback. *Optics Express* **25**, 2401–2412 (2017).
- [22] Vinckier, Q. *et al.* High-performance photonic reservoir computer based on a coherently driven passive cavity. *Optica* **2**, 438–446 (2015).

- [23] Argyris, A., Bueno, J. & Fischer, I. Pam-4 transmission at 1550 nm using photonic reservoir computing post-processing. *IEEE Access* **7**, 37017–37025 (2019).
- [24] Farmer, J. D. Chaotic attractors of an infinite-dimensional dynamical system. *Physica D* **4**, 366–393 (1982).
- [25] Le Berre, M. *et al.* Conjecture on the dimensions of chaotic attractors of delayed-feedback dynamical systems. *Physical Review A* **35**, 4020–4022 (1987).
- [26] Diekmann, O., Verduyn Lunel, S. M., van Gils, S. A. & Walther, H.-O. *Delay Equations* (Springer, New York, 1995).
- [27] Wu, J. *Introduction to Neural Dynamics and Signal Transmission Delay* (Walter de Gruyter, Berlin, Boston, 2001).
- [28] Erneux, T. *Applied Delay Differential Equations* (Springer, New York, 2009).
- [29] Atay, F. M. (ed.) *Complex Time-Delay Systems* (Springer, Berlin, 2010).
- [30] Michiels, W. & Niculescu, S.-I. *Stability, Control, and Computation for Time-Delay Systems* (Society for Industrial and Applied Mathematics, Philadelphia, 2014).
- [31] Erneux, T., Javaloyes, J., Wolfrum, M. & Yanchuk, S. Introduction to Focus Issue: Time-delay dynamics. *Chaos: An Interdisciplinary Journal of Nonlinear Science* **27**, 114201 (2017).
- [32] Yanchuk, S. & Giacomelli, G. Spatio-temporal phenomena in complex systems with time delays. *Journal of Physics A: Mathematical and Theoretical* **50**, 103001 (2017).
- [33] Paquot, Y. *et al.* Optoelectronic reservoir computing. *Scientific Reports* **2**, 287 (2012).
- [34] Schumacher, J., Toutounji, H. & Pipa, G. An analytical approach to single node delay-coupled reservoir computing. In *Artificial Neural Networks and Machine Learning – ICANN 2013*, 26–33 (Springer Berlin Heidelberg, Berlin, Heidelberg, 2013).
- [35] Stelzer, F., Röhm, A., Lüdge, K. & Yanchuk, S. Performance boost of time-delay reservoir computing by non-resonant clock cycle. *Neural Networks* **124**, 158–169 (2020).
- [36] Werbos, P. J. Applications of advances in nonlinear sensitivity analysis. In *System Modeling and Optimization: Proceedings of the 10th IFIP Conference*, 762–770 (Springer, Berlin, Heidelberg, 1982).
- [37] Rumelhart, D., Hinton, G. E. & Williams, R. J. Learning representations by back-propagating errors. *Nature* **323**, 533–536 (1986).
- [38] Mocanu, D., Mocanu, E., Stone, P. & et al. Scalable training of artificial neural networks with adaptive sparse connectivity inspired by network science. *Nat Commun* **9**, 2383 (2018).
- [39] Ardakani, A., Condo, C. & Gross, W. J. Sparsely-connected neural networks: Towards efficient VLSI implementation of deep neural networks. In *5th International Conference on Learning Representations, Conference Track Proceedings* (OpenReview.net, 2017).
- [40] Lecun, Y., Bottou, L., Bengio, Y. & Haffner, P. Gradient-based learning applied to document recognition. *Proceedings of the IEEE* **86**, 2278–2324 (1998).

- [41] Xiao, H., Rasul, K. & Vollgraf, R. Fashion-mnist: a novel image dataset for benchmarking machine learning algorithms. Preprint at <https://arxiv.org/abs/1708.07747> (2017).
- [42] Krizhevsky, A. Learning multiple layers of features from tiny images. *University of Toronto* (2012).
- [43] Netzer, Y. *et al.* Reading digits in natural images with unsupervised feature learning. *NIPS* (2011).
- [44] Giacomelli, G. & Politi, A. Relationship between Delayed and Spatially Extended Dynamical Systems. *Physical Review Letters* **76**, 2686–2689 (1996).
- [45] Hart, J. D., Schmadel, D. C., Murphy, T. E. & Roy, R. Experiments with arbitrary networks in time-multiplexed delay systems. *Chaos: An Interdisciplinary Journal of Nonlinear Science* **27**, 121103 (2017).
- [46] Hart, J. D., Larger, L., Murphy, T. E. & Roy, R. Delayed dynamical systems: networks, chimeras and reservoir computing. *Philosophical Transactions of the Royal Society A: Mathematical, Physical and Engineering Sciences* **377**, 20180123 (2019).
- [47] Sieber, J., Engelborghs, K., Luzyanina, T., Samaey, G. & Roose, D. Dde-biftool manual - bifurcation analysis of delay differential equations. Preprint at <https://arxiv.org/abs/1406.7144> (2016).
- [48] Breda, D., Diekmann, O., Gyllenberg, M., Scarabel, F. & Vermiglio, R. Pseudospectral discretization of nonlinear delay equations: New prospects for numerical bifurcation analysis. *SIAM Journal on Applied Dynamical Systems* **15**, 1–23 (2016).
- [49] Haber, E. & Ruthotto, L. Stable architectures for deep neural networks. *Inverse Problems* **34**, 014004 (2018).
- [50] Chen, R. T. Q., Rubanova, Y., Bettencourt, J. & Duvenaud, D. Neural ordinary differential equations. In *Proceedings of the 32nd International Conference on Neural Information Processing Systems*, 6572–6583 (Curran Associates Inc., Red Hook, NY, USA, 2018).
- [51] Lu, Y., Zhong, A., Li, Q. & Dong, B. Beyond finite layer neural networks: Bridging deep architectures and numerical differential equations. In *6th International Conference on Learning Representations, ICLR 2018 - Workshop Track Proceedings*, 3276–3285 (PMLR, Stockholm, Sweden, 2018).
- [52] Van der Sande, G., Brunner, D. & Soriano, M. C. Advances in photonic reservoir computing. *Nanophotonics* **6**, 561–576 (2017).
- [53] Tanaka, G. *et al.* Recent advances in physical reservoir computing: A review. *Neural Networks* **115**, 100–123 (2019).
- [54] Bishop, C. M. *Pattern Recognition and Machine Learning* (Springer, New York, 2006).
- [55] Srivastava, N., Hinton, G., Krizhevsky, A., Sutskever, I. & Salakhutdinov, R. Dropout: A simple way to prevent neural networks from overfitting. *Journal of Machine Learning Research* **15**, 1929–1958 (2014).
- [56] Glorot, X. & Bengio, Y. Understanding the difficulty of training deep feedforward neural networks. *Journal of Machine Learning Research - Proceedings Track* **9**, 249–256 (2010).

~~Deep Learning with a Single Neuron: Folding a Deep Neural Network in Time using Feedback-Modulated Delay Loops~~ Deep Neural Networks using a Single Neuron: Folded-in-Time Architecture using Feedback-Modulated Delay Loops

Supplementary Information

Florian Stelzer^{1,2,4}, André Röhm³, Raul Vicente⁴, Ingo Fischer³, and Serhiy Yanchuk¹

¹Institute of Mathematics, Technische Universität Berlin, 10623, Germany

²Department of Mathematics, Humboldt-Universität zu Berlin, 12489, Germany

³Instituto de Física Interdisciplinar y Sistemas Complejos, IFISC (UIB-CSIC), Campus Universitat de les Illes Balears, E-07122 Palma de Mallorca, Spain

⁴Institute of Computer Science, University of Tartu, Tartu, Estonia

1 Fit-DNN performance and confusion matrices for different number of hidden layers, choice of delays, and activation functions

Table 1 shows how the number of hidden layers L affects the performance of the Fit-DNN. We investigated two cases: the map limit $\theta \rightarrow \infty$ and the case $\theta = 0.5$. If the system operates in the map limit, we observe that the optimal number of hidden layers is 2 or 3, depending on the task. If $\theta = 0.5$, the performance of the Fit-DNN drops significantly for the CIFAR-10 [1], the coarse CIFAR-100 [1], the cropped SVHN [2], and the denoising task. For this reason, deeper networks do not offer an advantage for solving these tasks if $\theta = 0.5$. The MNIST [3] and Fashion-MNIST [4] accuracies do not suffer much from choosing a small node separation θ . Here the systems performance remains almost unchanged in comparison to the map limit.

Figure 1 shows the effect of the choice of the number of delays D on the performance of the Fit-DNN. A larger number of delays D yields a slightly better accuracy for the CIFAR-10 task. We obtain an accuracy of less than 51% for $D = 25$, and an accuracy between 52% and 53% for $D = 125$ or larger. For the denoising task, we already obtain a good mean squared error (MSE) for a small number of delays D . The MSE remains mostly between 0.0253 and 0.0258 independently of D . The fluctuations of the MSE are small.

We ~~compare~~ compared two methods for choosing the delays $\tau_d = n_d \theta$. The first method is to draw the numbers n_d without replacement from a uniform distribution on the set $\{1, \dots, 2N - 1\}$. The second method is to choose equidistant delays, with $n_{d+1} - n_d = \lfloor (2N - 1)/D \rfloor$. For the CIFAR-10 task, one may observe a slight advantage of the equidistant delays, whereas for the denoising task, randomly chosen delays yield slightly better results. In both cases, however, the influence of the chosen method on the quality of the results is small and seems to be insignificant.

Table 2 compares the performance of the Fit-DNN for different activation functions $f(a) = \sin(a)$, $f(a) = \tanh(a)$, and $f(a) = \max(0, a)$ (ReLU). The results show that the Fit-DNN works well with various activation functions.

Figure 2 shows the confusion matrices for the cropped SVHN and the CIFAR-10 tasks. These matrices show how often images of a corresponding dataset class are either recognized correctly or mismatched with another class. Confusion matrices are a suitable tool to identify which classes are confused more or less often. The confusion matrix for the cropped SVHN task shows, e.g., that the number 3 is relatively often falsely recognized as 5 or 9, but almost never as 4 or 6. The confusion matrix for the CIFAR-10 task indicates that images from animal classes (bird, cat, deer, dog, frog, horse) are often mismatched with another animal class, but rarely with a transportation class (airplane, automobile, ship, truck). This is an expected result for the CIFAR-10 task.

Figure 3 shows results for a sine function fitting task. The objective of the task is to fit functions $y_i(u)$, $i = 1, \dots, 5$, $u \in [-1, 1]$, plotted in Fig. 4, which are defined as concatenations $y_i(u) = s_i \circ \dots \circ s_1(u)$ of sine functions $s_i(u) = \sin(\omega_i(u) + \varphi_i)$ with

$$\omega_1 = 0.65 \cdot 2\pi, \quad \omega_2 = 0.4 \cdot 2\pi, \quad \omega_3 = 0.3 \cdot 2\pi, \quad \omega_4 = 0.55 \cdot 2\pi, \quad \omega_5 = 0.45 \cdot 2\pi, \quad (1)$$

$$\varphi_1 = 1.0, \quad \varphi_2 = -0.5, \quad \varphi_3 = -0.3, \quad \varphi_4 = 0.6, \quad \varphi_5 = 0.2. \quad (2)$$

The simulations were performed with $N = 20$ nodes per hidden layer, $D = 3$, and $\tau_1 = 15$, $\tau_2 = 20$, $\tau_3 = 25$. Since the task is to fit a concatenation of i sine functions and the Fit-DNN consists in this case of L concatenated sine functions, one would expect optimal results for $L \geq i$. In our tests, this was true for up to $i = 3$ concatenated functions. The function y_1 can be approximated by the Fit-DNN's output with a small MSE with any number of layers, see Fig. 3. The function y_2 can be fitted with a small error if and only if $L \geq 2$ (with a few exceptions). For the function y_3 we obtain relatively exact approximations with 2 or more hidden layers, but the smallest MSE is obtained with $L = 3$ in most cases. The Fit-DNN fails to fit the functions y_4 and y_5 for all L .

L	1	2	3	4		
$\theta = 0.5$	MNIST	98.43	98.54	98.3	98.24	[%]
	Fashion-MNIST	87.61	87.87	87.51	87.44	[%]
	CIFAR-10	52.35	52.13	52.05	51.32	[%]
	coarse CIFAR-100	33.52	33.22	32.51	31.32	[%]
	cropped SVHN	78.26	78.78	78.21	78.39	[%]
	denoising ($D = 5$)	0.0250	0.0254	0.0269	0.0362	[MSE]
	denoising ($D = 50$)	0.0251	0.0253	0.0269	0.0278	[MSE]
map limit	MNIST	98.41	98.62	98.47	98.58	[%]
	Fashion-MNIST	87.22	87.91	87.97	87.88	[%]
	CIFAR-10	53.69	54.57	54.28	54.15	[%]
	coarse CIFAR-100	35.13	35.69	35.77	36.48	[%]
	cropped SVHN	80.38	82.71	82.92	82.22	[%]
	denoising ($D = 5$)	0.0255	0.0244	0.0246	0.0250	[MSE]
	denoising ($D = 50$)	0.0257	0.0241	0.0243	0.0246	[MSE]

Table 1: Accuracies [%] for the classification tasks and mean squared error for the denoising task for different numbers of hidden layers L . For a node separation of $\theta = 0.5$, two hidden layers seem to be optimal for the classification tasks (except CIFAR-10/100), and one hidden layer is sufficient for the denoising task. When the systems operates in the map limit $\theta \rightarrow \infty$, additional hidden layers can improve the performance.

Figure 1: Accuracy and MSE for different numbers of delays D . For each D , the plots show 5 results (cross symbols) with delays drawn from a uniform distribution (blue), and equidistant delays (red). The circle-dot symbols connected by solid lines show the mean of the results.

f	sin	tanh	ReLU	
MNIST	98.396	98.611	98.833	[%]
Fashion-MNIST	87.921	88.78	89.177	[%]
CIFAR-10	52.062	51.039	50.598	[%]
coarse CIFAR-100	32.162	32.183	30.737	[%]
cropped SVHN	78.292	77.739	77.916	[%]
denoising	0.0254	0.0249	0.0255	[MSE]

Table 2: Accuracies [%] for the classification tasks and mean squared error for the denoising task for different activation functions f . Overall, the compared activation functions work similarly well.

Figure 2: Numbers of images from the cropped SVHN and CIFAR-10 test sets by their actual class and the Fit-DNN's prediction. The CIFAR-10 confusion matrix implies that false predictions occur mostly within the superclasses *animals* and *transportation* but rarely between the superclasses.

Figure 3: The plot shows the mean squared errors (MSE) for fitting the functions $y_i(u)$, $i = 1, \dots, 5$ with different numbers of layers L . We repeated the numerical experiment five times, each panel shows the results of one of these independent repetitions.

Figure 4: Functions $y_i(u)$, $i = 1, \dots, 5$.

Figure 5: ~~Solution Sketch of the solution~~ of delay system (3)–(4) with delay-induced connections. Red arrows correspond to ~~the~~ a delay $0 < \tau_1 < T$, and yellow to $T < \tau_2 < 2T$. Dashed lines with symbol \times indicate ~~the removed connections due to~~ that were removed by setting the ~~vanishing~~ modulation amplitude to zero; see Eq. (6).

2 The Fit-DNN delay system and ~~map network~~ representation

2.1 Generating delay system

The Fit-DNN has M input nodes, P output nodes, and L hidden layers, each consisting of N nodes. The hidden layers are described by the delay system

$$\dot{x}(t) = -\alpha x(t) + f(a(t)), \quad (3)$$

$$a(t) = J(t) + b(t) + \sum_{d=1}^D \mathcal{M}_d(t)x(t - \tau_d), \quad (4)$$

where $\alpha > 0$ is a constant time-scale, f is a nonlinear activation function, and the argument $a(t)$ is a signal ~~which is~~ composed of a data signal $J(t)$, a bias signal $b(t)$, and delayed feedback terms modulated by functions $\mathcal{M}_d(t)$. The components of $a(t)$ are described in the Methods Section. The delays are given by $\tau_d = n_d\theta$, where $\theta := T/N$ and $1 \leq n_1 < \dots < n_D \leq 2N - 1$ are natural numbers. The state of the ℓ -th hidden layer is given by the solution $x(t)$ of (3)–(4) on the interval $(\ell - 1)T < t \leq \ell T$. We define the node states of the hidden layers as follows:

$$x_n^\ell := x((\ell - 1)T + n\theta) \quad (5)$$

for the node $n = 1, \dots, N$ of ~~the~~ layer $\ell = 1, \dots, L$.

The nodes of the hidden layers are connected by the delays τ_d , ~~which is as~~ illustrated in Fig. 5. ~~In order to~~ To ensure that only nodes of consecutive hidden layers are connected, we set

$$\mathcal{M}_d(t) = 0 \quad \text{if } t \in ((\ell - 1)T, \ell T] \text{ and } t - \tau_d = t - n_d\theta \notin ((\ell - 2)T, (\ell - 1)T]. \quad (6)$$

The delay connections, which are set to zero by condition (6), are indicated by dashed arrows marked with a black \times symbol in Fig. 5.

Additionally, we set $\mathcal{M}_d(t) = 0$ for $t \in [0, T]$. This implies, in combination with condition (6), that the system has no incoming delay ~~connection connections~~ from a time $t - \tau_d$ before zero. For this reason, ~~we do not need to know~~ a history function [5, 6, 7, 8] ~~for is not required to solve~~ the delay system (3)–(4) ~~to compute the solution~~ for positive time. Knowing the initial condition $x(0) = x_0$ at a single point is sufficient.

System (3)–(4) is defined on the interval $[0, LT]$. ~~Applying The application of~~ the variation of constants formula ~~yields gives~~ for $0 \leq t_0 < t \leq LT$ the equation

$$x(t) = e^{-\alpha(t-t_0)}x(t_0) + \int_{t_0}^t e^{\alpha(s-t)}f(a(s))ds. \quad (7)$$

Using this equation on appropriate time intervals $[(n - 1)\theta, n\theta]$, we obtain the following relations for the nodes in the first hidden layer

$$x_1^1 = e^{-\alpha\theta}x_0 + \int_0^\theta e^{\alpha(s-\theta)}f(a(s))ds, \quad (8)$$

$$x_n^1 = e^{-\alpha\theta}x_{n-1}^1 + \int_0^\theta e^{\alpha(s-\theta)}f(a((n-1)\theta + s))ds, \quad n = 2, \dots, N. \quad (9)$$

Here $x_0 = x(0)$ is the initial state of system (3)–(4). Similarly, for the hidden layers $\ell = 2, \dots, L$, we have

$$x_1^\ell = e^{-\alpha\theta} x_N^{\ell-1} + \int_0^\theta e^{\alpha(s-\theta)} f(a((\ell-1)T+s)) ds, \quad (10)$$

$$x_n^\ell = e^{-\alpha\theta} x_{n-1}^\ell + \int_0^\theta e^{\alpha(s-\theta)} f(a((\ell-1)T+(n-1)\theta+s)) ds, \quad n = 2, \dots, N. \quad (11)$$

For the first hidden layer, the signal $a(t)$ is piecewise constant. More specifically,

$$a(s) = J(s) = a_n^1 = g(a_n^{\text{in}}), \quad (n-1)\theta < s \leq n\theta, \quad n = 1, \dots, N, \quad (12)$$

where

$$a^{\text{in}} = w_{n,M+1}^{\text{in}} + \sum_{m=1}^M w_{nm}^{\text{in}} u_m. \quad (13)$$

Taking into account Eq. (12), relations (8)–(9) lead to the following exact expressions for the nodes of the first hidden layer:

$$x_1^1 = e^{-\alpha\theta} x_0 + \alpha^{-1}(1 - e^{-\alpha\theta})f(a_1^1), \quad (14)$$

$$x_n^1 = e^{-\alpha\theta} x_{n-1}^1 + \alpha^{-1}(1 - e^{-\alpha\theta})f(a_n^1), \quad n = 2, \dots, N. \quad (15)$$

2.2 **Map-Network** representation for small node separations

For the hidden layers $\ell = 2, \dots, L$, i.e., for $T < t \leq LT$, the signal $a(t)$ is defined by

$$a(t) = b(t) + \sum_{d=1}^D \mathcal{M}_d(t)x(t - \tau_d), \quad (16)$$

where $b(t)$ and $\mathcal{M}_d(t)$ are piecewise constant functions with discontinuities at the grid points $n\theta$. However, the feedback signals $x(t - \tau_d)$ are not piecewise constant. Therefore, we cannot replace $a((\ell-1)T + (n-1)\theta + s)$, $0 < s < \theta$, in Eq. (10) and (11) by constants. However, if the node separation θ is small, we can approximate the value of

$$x((\ell-1)T + (n-1)\theta + s - \tau_d) = x((\ell-1)T + (n - n_d - 1)\theta + s), \quad 0 < s < \theta, \quad (17)$$

by the value $x((\ell-1)T + n\theta - \tau_d)$, which can be rewritten as

$$x((\ell-1)T + n\theta - \tau_d) = x((\ell-1)T + (n - n_d)\theta) = x((\ell-2)T + (n - n'_d)\theta), \quad (18)$$

where $n'_d = n_d - N$. Condition (6) ensures that the layer $\ell - 1$ is connected only to the previous layer $\ell - 2$. Formally, it means the nonzero values $v_{d,n}^\ell$ of \mathcal{M}_d allow only such connections that $1 \leq n - n'_d \leq N$, i.e., $x((\ell-2)T + (n - n'_d)\theta) = x_{n-n'_d}^{\ell-2}$. As a result, we can approximate $a((\ell-1)T + (n-1)\theta + s)$ for $0 < s < \theta$ by

$$a_n^\ell = w_{n,N+1}^\ell + \sum_{j=1}^N w_{nj}^\ell x_j^{\ell-1}, \quad n = 1, \dots, N, \quad \ell = 2, \dots, L, \quad (19)$$

where

$$w_{nj}^\ell := \delta_{N+1,j} b_n^\ell + \sum_{d=1}^D \delta_{n-n'_d,j} v_{d,n}^\ell \quad (20)$$

defines a weight matrix $W^\ell = (w_{nj}^\ell) \in \mathbb{R}^{N \times (N+1)}$ for the connections from layer $\ell - 1$ to layer ℓ . This matrix is illustrated in Fig. 6.

In summary, we obtain the following **map-network** representation of the Fit-DNN, illustrated in Fig. 7, which approximates the **nodes-node** states up to first order terms in θ . The first hidden layer is given by

$$x_1^1 = e^{-\alpha\theta} x_0 + \alpha^{-1}(1 - e^{-\alpha\theta})f(a_1^1), \quad (21)$$

$$x_n^1 = e^{-\alpha\theta} x_{n-1}^1 + \alpha^{-1}(1 - e^{-\alpha\theta})f(a_n^1), \quad n = 2, \dots, N. \quad (22)$$

The hidden layers $\ell = 2, \dots, L$ are given by

$$x_1^\ell = e^{-\alpha\theta} x_N^{\ell-1} + \alpha^{-1}(1 - e^{-\alpha\theta})f(a_1^\ell), \quad (23)$$

$$x_n^\ell = e^{-\alpha\theta} x_{n-1}^\ell + \alpha^{-1}(1 - e^{-\alpha\theta})f(a_n^\ell), \quad n = 2, \dots, N, \quad (24)$$

and the output layer is defined by

$$\hat{y}_p := f_p^{\text{out}}(a^{\text{out}}), \quad p = 1, \dots, P, \quad (25)$$

Figure 6: Illustration of the sparse weight matrix W^ℓ containing the connection weights between the hidden layers $\ell - 1$ and ℓ , see Eq. (20). The nonzero weights are arranged on diagonals, and equal to the values $v_{d,n}^\ell$ of the functions \mathcal{M}_d . The position of the diagonals is determined by the corresponding delays τ_d . If $\tau_d = N\theta = T$, then the main diagonal contains the entries $v_{d,1}^\ell, \dots, v_{d,N}^\ell$. If $\tau_d = n_d\theta < T$, then the corresponding diagonal lies above the main diagonal and contains the values $v_{d,1}^\ell, \dots, v_{d,n_d}^\ell$. ~~If $\tau_d = n_d\theta > T$, then the corresponding diagonal lies below the main diagonal and contains the values $v_{d,n'_d+1}^\ell, \dots, v_{d,N}^\ell$, where $n'_d = n_d - N$.~~ The last column of the matrix contains the bias weights.

Figure 7: The multilayer neural network described by the equations (21)–(28). ~~The Adaptable connection weights which are subject to training are plotted in green. The connection weights connections between the input layer and the first hidden layer as well as the weights connections between the last hidden and the output layer are dense (all-to-all connection). The connections between the hidden layers are sparse. See in general sparsely connected; see Fig. 6 for an illustration of the connection matrices between the hidden layers. The difference In contrast to a classical multilayer perceptron are, the Fit-DNN comprises fixed linear connections between the neighboring nodes, which are plotted in (black arrows). These additional connections make it necessary to use a modified back-propagation method to compute must be taken into account when computing the error gradients of the network. Note that also the hidden layers, specifically namely the nodes $x_{N-1}^{\ell-1}$ and x_1^ℓ , are also directly linked connected by this type of additional such linear connections. Thereby, the activation vectors $(a_1^\ell, \dots, a_N^\ell)^T$ are skipped links.~~

where f^{out} is an output activation function which suits to the given task. Moreover,

$$a_n^1 := g(a_n^{\text{in}}) := g\left(\sum_{m=1}^{M+1} w_{nm}^{\text{in}} u_m\right), \quad n = 1, \dots, N, \quad (26)$$

$$a_n^\ell := \sum_{j=1}^{N+1} w_{nj}^\ell x_j^{\ell-1}, \quad n = 1, \dots, N, \quad \ell = 2, \dots, L, \quad (27)$$

$$a_p^{\text{out}} := \sum_{n=1}^{N+1} w_{pn}^{\text{out}} x_n^L, \quad p = 1, \dots, P, \quad (28)$$

where $u_{M+1} := 1$ and $x_{N+1}^\ell := 1$, for $\ell = 1, \dots, L$. We call the a_n^ℓ and a_p^{out} the activation of the corresponding node. For $n = 1, \dots, N$, the variable w_{pn}^{out} denotes the output weight connecting the n -th node of layer L to the p -th output node, and $w_{p,(N+1)}^{\text{out}}$ denotes the bias for p -th output node (in other words, the weight connecting the on-neuron x_{N+1}^L of layer L to the p -th output node).

The topology of the obtained map network representation of the Fit-DNN does not depend on the discretization choice method. Instead of following the above derivation, one could simply approximate the node states simply by applying an Euler scheme to the delay system (3)–(4). Then one obtains the The obtained map

$$x_n^\ell = x_{n-1}^\ell + \theta f(a_n^\ell), \quad (29)$$

which possesses the same connections as the map network representation (21)–(28) of the Fit-DNN, but has slightly different connection weights. Nonetheless Nevertheless, for our purposes it is necessary to consider (21)–(28) instead of the simple Euler scheme (29). The weights $e^{-\alpha\theta}$ of the linear connections of neighboring nodes in EqEqs. (21)–(24) are only slightly smaller than the corresponding weights 1 in Eq. (29), but the indirect connection from \$x_n^\ell\$ to a distant node \$x_\nu^\ell\$ with \$\nu \gg n\$ is \$e^{-(\nu-n)\alpha\theta} \approx 0\$. This fact helps us to calculate they allow to avoid destabilization during the computation of the error gradient of the Fit-DNN by the modified back-propagation algorithm derived in the section below. Using the simple Euler scheme, the connections weights between distant nodes remain to be 1, and lead to accurate results.

2.3 Map limit

Here we show that the nodes of the Fit-DNN (8)–(11) can be approximated by the map limit

$$x_n^\ell = \alpha^{-1} f(a_n^\ell) \quad (30)$$

for large node separation θ , up to exponentially small terms $\mathcal{O}(e^{-\beta\theta})$ for all $0 < \beta < \alpha$. This limit corresponds to the approach for building networks of coupled maps from delay systems in [9, 10].

For the nodes of the first hidden layer, EqEqs. (14)–(15) provide exact solutions for any θ . Hence, replacing θ by $r \in [0, \theta]$, we obtain for the values of $x(t)$ in the interval $[(n-1)\theta, n\theta]$

$$x((n-1)\theta + r) = e^{-\alpha r} x_{n-1}^1 + \alpha^{-1} (1 - e^{-\alpha r}) f(a_n^1), \quad (31)$$

which implies that the solution $x(t)$ decays exponentially to $\alpha^{-1} f(a_n^1)$. In other words, it holds

$$x((n-1)\theta + r) = \alpha^{-1} f(a_n^1) + \mathcal{O}(e^{-\alpha r}). \quad (32)$$

In order to To show similar exponential estimates for the layers $\ell = 2, \dots, L$, we use an inductive argument inductive arguments. For this, we assume that the following estimate holds for layer $\ell - 1$:

$$x((\ell-2)T + (n-1)\theta + r) = \alpha^{-1} f(a_n^{\ell-1}) + \mathcal{O}(e^{-\beta r}) \quad (33)$$

for all $0 < \beta < \alpha$, $r \in [0, \theta]$, and all n within the layer. Note that this estimate is true for the first hidden layer because (33) is a weaker statement than (32). For layer ℓ , we obtain from Eq. (7)

$$x((\ell-1)T + (n-1)\theta + r) = e^{-\alpha r} x_{n-1}^\ell + \int_0^r e^{\alpha(s-r)} f(a((\ell-1)T + (n-1)\theta + s)) ds, \quad (34)$$

where (33) implies

$$\underline{a((\ell-1)T + (n-1)\theta + s)} = b((\ell-1)T + (n-1)\theta + s) + \sum_{d=1}^D \mathcal{M}_d^\ell((\ell-1)T + (n-1)\theta + s) x((\ell-1)T + (n-1)\theta + s - \tau_d) = b_n^\ell + \sum_{d=1}^D v_{d,n}^\ell x$$

(35)

We obtain the term $\mathcal{O}(e^{-\beta s})$ in Eq. (35) because Eq. (33) implies

$$x((\ell - 2)T + (n - 1)\theta - n'_d\theta + s) + \mathcal{O}(e^{-\beta s}) = \alpha^{-1}f(a_{n-n'_d}^{\ell-1}) = x((\ell - 2)T - (n - n'_d)\theta) + \mathcal{O}(e^{-\beta\theta}) \quad (36)$$

and $e^{-\beta\theta} < e^{-\beta s}$. If f is Lipschitz continuous (which is the case for all our examples), it follows from Eqs. (34) and (35) that

$$x((\ell - 1)T + (n - 1)\theta + r) = e^{-\alpha r}x_{n-1}^\ell + \alpha^{-1}(1 - e^{-\alpha r})f(a_n^\ell) + \int_0^r e^{\alpha(s-r)}\mathcal{O}(e^{-\beta s})ds. \quad (37)$$

Since

$$\int_0^r e^{\alpha(s-r)}e^{-\beta s}ds = \frac{1}{\alpha - \beta}(e^{-\beta r} - e^{-\alpha r}) < \frac{e^{-\beta r}}{\alpha - \beta}, \quad (38)$$

we obtain

$$x((\ell - 1)T + (n - 1)\theta + r) = \alpha^{-1}f(a_n^\ell) + \mathcal{O}(e^{-\beta r}). \quad (39)$$

This ~~hold holds~~ in particular for $r = \theta$. Therefore, we have shown that Eq. (30) holds up to terms of order $\mathcal{O}(e^{-\beta\theta})$ for all $0 < \beta < \alpha$.

3 Back-propagation for the Fit-DNN

~~The map~~ To calculate the error gradient of a traditional multilayer perceptron, it sufficient to compute partial derivatives of the loss function with respect to the node activations $\partial\mathcal{E}/\partial a_n^\ell$ by an iterative application of the chain rule and to store them as intermediate results. These derivatives are called *error signals* and are denoted by δ_n^ℓ . Subsequently, the weight gradient can be calculated by applying the chain rule again for each weight, i.e., the *back-propagation*.

~~The network~~ representation (21)–(28) of the Fit-DNN, illustrated by Fig. 7, contains additional linear connections which ~~do not occur are not present~~ in classical multilayer perceptrons. ~~Thus, we cannot apply the classical back-propagation algorithm for the gradient computation, which is necessary for the weight training. Instead, we derive a modified back-propagation algorithm which is specific for—~~

~~Firstly, we define extended activation vectors—~~

$$\underline{\bar{a}^\ell} := \begin{pmatrix} a_1^\ell \\ \vdots \\ a_N^\ell \\ a_{N+1}^\ell \end{pmatrix} \in \mathbb{R}^{N+1},$$

for $\ell = 1, \dots, L$, where $\bar{a}_{N+1}^\ell := x_{N+1}^{\ell-1}$, for $\ell = 2, \dots, L$, and $\bar{a}_{N+1}^1 := x_0$. Moreover, we introduce the following notations for the partial derivatives of the loss function w.r.t. the activation vectors. ~~We need to take these connections into account when calculating the weight gradient of the loss function, more specifically, the so-called error signals—~~

$$\begin{aligned} \underline{\Delta_p^{\text{out}}} &:= \partial_p \mathcal{E}(a^{\text{out}}), & \text{for } p = 1, \dots, P, \\ \underline{\Delta_n^\ell} &:= \partial_n \mathcal{E}(\bar{a}^\ell), & \text{for } n = 1, \dots, N+1, \ell = 1, \dots, L. \end{aligned}$$

error signals δ_n^ℓ . Despite having these additional connections, all nodes are still strictly forward-connected. Consequently, we can calculate the error signals by applying the chain rule node by node. Thereby, we employ a second type of error signal $\underline{\Delta_n^\ell} := \partial\mathcal{E}/\partial a_n^\ell$ because the local connections (black arrows in Fig. 7) do not enter the nodes through the activation function. Thus, we need to know $\underline{\Delta_n^\ell}$ for the back-propagation via these local connections. However, memory efficient implementations are possible because we only need to store one $\underline{\Delta_n^\ell}$ at a time. The weight gradient can again be calculated from the error signals δ_n^ℓ by using the chain rule once more for each weight.

The ~~gradient~~ back-propagation algorithm for the Fit-DNN is described in the Methods Section. In the following we explain the Steps 1–4 of this algorithm in detail.

Step 1: ~~For certain favorable choices~~ of the loss function ~~for the map representation—~~ of the Fit-DNN with respect to the input, hidden, and output weights can be calculated using the following formulas: We have $\underline{\Delta_p^{\text{out}}} = \hat{y}_p - y_p$ if f^{out} is the softmax function and \mathcal{E} is the cross-entropy loss, or and the output activation function f^{out} is the identity function and \mathcal{E} is the mean-square loss. ~~The error signals—~~, we can compute the error signal of the output layer by the following simple equation:

$$\underline{\delta_p^{\text{out}}} = \frac{\partial\mathcal{E}}{\partial a_p^{\text{out}}} = \hat{y}_p - y_p, \quad (40)$$

for $p = 1, \dots, P$. This holds in particular for combining the cross-entropy loss function with the softmax output function and for combining the mean-squared loss function with the identity output function. For a derivation we refer to [11] or [12].

Step 2: The formulas for the error signals of the last hidden layer are

$$\underline{\Delta_{N+1}^L} = \sum_{p=1}^P \Delta_p^{\text{out}} \sum_{j=1}^N e^{-\alpha\theta j} w_{pj}^{\text{out}},$$

$$\underline{\Delta_n^L} = \Phi f'(a_n^L) \sum_{p=1}^P \Delta_p^{\text{out}} \sum_{j=n}^N e^{-\alpha\theta(j-n)} w_{pj}^{\text{out}}, \quad \text{for } n = 1, \dots, N,$$

where $\Phi := \alpha^{-1}(1 - e^{-\alpha\theta})$. The error signals for the remaining hidden layers $\ell = L-1, \dots, 1$ are

$$\underline{\Delta_{N+1}^\ell} = e^{-\alpha\theta N} \Delta_{N+1}^{\ell+1} + \sum_{i=1}^N \Delta_i^{\ell+1} \sum_{j=1}^N e^{-\alpha\theta j} w_{ij}^{\ell+1},$$

$$\underline{\Delta_n^\ell} = \Phi f'(a_n^\ell) \left[e^{-\alpha\theta(N-n)} \Delta_{N+1}^{\ell+1} + \sum_{i=1}^N \Delta_i^{\ell+1} \sum_{j=n}^N e^{-\alpha\theta(j-n)} w_{ij}^{\ell+1} \right],$$

for $n = 1, \dots, N$. The partial derivatives of the loss function with respect to the output weights are

$$\underline{\frac{\partial \mathcal{E}}{\partial w_{pn}^{\text{out}}}} = \Delta_p^{\text{out}} x_n^L, \quad \text{for } n = 1, \dots, N+1, p = 1, \dots, P.$$

The partial derivatives with respect to the hidden weights are can be found by applying the chain rule twice. Let $\Phi := \alpha^{-1}(1 - e^{-\alpha\theta})$. The error derivatives w.r.t. the node states of the last hidden layer can be calculated from the output error signals and the output weight. We have

$$\underline{\frac{\partial \mathcal{E}}{\partial w_{nj}^\ell}} = \Delta_n^\ell x_j^{\ell-1}, \text{ for } j_N^L = 1, \dots, N+1, n \underline{\frac{\partial \mathcal{E}}{\partial x_N^L}} = 1, \dots, \sum_{p=1}^P \frac{\partial \mathcal{E}}{\partial a_p^{\text{out}}} \frac{\partial a_p^{\text{out}}}{\partial x_N^L} = \sum_{p=1}^P \delta_p^{\text{out}} w_{pN}^{\text{out}}, N. \quad (41)$$

The partial derivatives in the input weights are

$$\underline{\frac{\partial \mathcal{E}}{\partial w_{nm}^{\text{in}}}} = \Delta_n^1 g'(a_n^{\text{in}}) u_m, \quad \text{for } m = 1, \dots, M+1, n = 1, \dots, N.$$

For a proof of statement (i) see, e.g., reference [11]. The definition of Δ_p^{out} coincides with the definition of δ_p^{out} used for the formulation of the classical back-propagation algorithm.

In the following we give a technical proof of the statements (ii)-(viii).

(ii) The chain rule implies

$$\underline{\Delta_n^L} = \partial_n \mathcal{E}(\bar{a}^L) = \sum_{p=1}^P \partial_p \mathcal{E}(a^{\text{out}}) \partial_n a_p^{\text{out}}(\bar{a}^L) = \sum_{p=1}^P \Delta_p^{\text{out}} \partial_n a_p^{\text{out}}(\bar{a}^L),$$

for $n = 1, \dots, N+1$. By Eq. we have

$$\underline{a_p^{\text{out}}} = \sum_{j=1}^{N+1} w_{pj}^{\text{out}} x_j^L,$$

where $x_{N+1}^L = 1$ is constant. The other node states x_n^L depend on the vector \bar{a}^L . Applying Eq. and Eq. iteratively and inserting $\bar{a}_{N+1}^L = x_{N+1}^L = 1$ yields

$$\underline{x_j^L} = e^{-\alpha\theta j} \bar{a}_{N+1}^L + \Phi \sum_{\nu=1}^j e^{-\alpha\theta(j-\nu)} f(\bar{a}_\nu^L).$$

and

$$\underline{\Delta_n^L} = \frac{\partial \mathcal{E}}{\partial x_n^L} = \frac{\partial \mathcal{E}}{\partial x_{n+1}^L} \frac{\partial x_{n+1}^L}{\partial x_n^L} + \sum_{p=1}^P \frac{\partial \mathcal{E}}{\partial a_p^{\text{out}}} \frac{\partial a_p^{\text{out}}}{\partial x_n^L} = \Delta_{n+1}^L e^{-\alpha\theta} + \sum_{p=1}^P \delta_p^{\text{out}} w_{pn}^{\text{out}}, \quad (42)$$

Thus,

$$\underline{\partial_{N+1} a_p^{\text{out}}(\bar{a}^L)} = \sum_{j=1}^N w_{pj}^{\text{out}} \partial_{N+1} x_j^L(\bar{a}^L) = \sum_{j=1}^N w_{pj}^{\text{out}} e^{-\alpha\theta j}$$

for $n = N-1, \dots, 1$. The error derivatives w.r.t. the node activations can then be calculated by multiplication with the corresponding derivative of the activation function, i.e.,

$$\underline{\delta_n^L} = \frac{\partial \mathcal{E}}{\partial a_n^L} = \frac{\partial \mathcal{E}}{\partial x_n^L} \frac{\partial x_n^L}{\partial a_n^L} = \Delta_n^L \Phi f'(a_n^L), \quad (43)$$

and for $n = 1, \dots, N$ we have for $n = 1, \dots, N$.

Step 3: Also for the remaining hidden layers, we need only to apply the chain rule twice to obtain the formulas for the error signals. For $\ell = L-1, \dots, 1$, we have

$$\underline{\partial_n a_p^{\text{out}(L)} \Delta_N^\ell} = \frac{\partial \mathcal{E}}{\partial x_N^\ell} = \frac{\partial \mathcal{E}}{\partial x_1^{\ell+1}} \frac{\partial x_1^{\ell+1}}{\partial x_N^\ell} + \sum_{j=1, i=1}^N w_{pj}^{\text{out}} \partial_n x_j^L(L) \frac{\partial \mathcal{E}}{\partial a_i^{\ell+1}} \frac{\partial a_i^{\ell+1}}{\partial x_N^\ell} = \underline{\Delta_{i=1}^{\ell+1} e^{-\alpha\theta} + \sum_{j=n, i=1}^N \delta_i^{\ell+1} w_{pj}^{\text{out}} \Phi e^{-\alpha\theta(j-n)} f'(L)_{iN}^{\ell+1}}, \quad (44)$$

because

$$\underline{\partial_n x_j^L(\bar{a}^L)} = \begin{cases} 0, & \text{if } j < n, \\ \Phi e^{-\alpha\theta(j-n)} f'(\bar{a}_n^L), & \text{if } j \geq n. \end{cases}$$

Inserting Eq. into Eq. yields Eq. and inserting Eq. into Eq. yields Eq.

(iii) The chain rule implies and

$$\underline{\Delta_n^{\ell\ell}} = \underline{\partial_n \mathcal{E}(\ell) \frac{\partial \mathcal{E}}{\partial x_n^\ell}} = \underline{\partial_{N+1} \mathcal{E}(\ell+1) \partial_n^{\ell+1}(\ell)} \frac{\partial \mathcal{E}}{\partial x_{n+1}^\ell} \frac{\partial x_{n+1}^\ell}{\partial x_n^\ell} + \sum_{i=1}^N \underline{\partial_i \mathcal{E}(\ell+1) \partial_n^{\ell+1}(\ell)} \frac{\partial \mathcal{E}}{\partial a_i^{\ell+1}} \frac{\partial a_i^{\ell+1}}{\partial x_n^\ell} = \underline{\Delta_{N+1}^{\ell+1} \partial_n^{\ell+1}(\ell)_{n+1}^\ell e^{-\alpha\theta}} + \sum_{i=1}^N \underline{\Delta_i^{\ell+1} \partial_n^{\ell+1}(\ell)} \quad (45)$$

for $n = 1, \dots, N+1$. From Eq. and Eq. follows $n = N-1, \dots, 1$. Again, the error derivatives w.r.t. the node activations can be calculated by multiplication with the derivative of the activation function:

$$\underline{\delta_n^{\ell+1}} = \underline{x_N^\ell \frac{\partial \mathcal{E}}{\partial a_n^\ell}} = \underline{e^{-\alpha\theta N} \delta_{N+1}^\ell} + \underline{\frac{\partial \mathcal{E}}{\partial x_n^\ell} \frac{\partial x_n^\ell}{\partial a_n^\ell}} = \underline{\Delta_n^\ell \Phi \sum_{\nu=1}^N e^{-\alpha\theta(N-\nu)} f'(\nu a_n^\ell)}, \quad (46)$$

and hence

$$\underline{\partial_{N+1} \bar{a}_{N+1}^{\ell+1}(\bar{a}^\ell)} = \underline{e^{-\alpha\theta N}}$$

and

$$\underline{\partial_n \bar{a}_{N+1}^{\ell+1}(\bar{a}^\ell)} = \underline{\Phi e^{-\alpha\theta(N-n)} f'(\bar{a}_n^\ell)},$$

for $n = 1, \dots, N$. Furthermore, for $i = 1, \dots, N$, Eq. implies

$$\underline{\bar{a}_i^{\ell+1}} = \underline{\sum_{j=1}^{N+1} w_{ij}^{\ell+1} x_j^\ell},$$

where $x_{N+1}^\ell = 1$ is constant and applying Eqs. iteratively yields

$$\underline{x_j^\ell} = \underline{e^{-\alpha\theta j} \bar{a}_{N+1}^\ell + \Phi \sum_{\nu=1}^j e^{-\alpha\theta(j-\nu)} f(\bar{a}_\nu^\ell)}.$$

This implies

$$\underline{\partial_{N+1} \bar{a}_i^{\ell+1}(\bar{a}^\ell)} = \underline{\sum_{j=1}^N w_{ij}^{\ell+1} \partial_{N+1} x_j^\ell(\bar{a}^\ell)} = \underline{\sum_{j=1}^N w_{ij}^{\ell+1} e^{-\alpha\theta j}}$$

and

$$\underline{\partial_n \bar{a}_i^{\ell+1}(\bar{a}^\ell)} = \underline{\sum_{j=1}^N w_{ij}^{\ell+1} \partial_n x_j^\ell(\bar{a}^\ell)} = \underline{\sum_{j=n}^N w_{ij}^{\ell+1} \Phi e^{-\alpha\theta(j-n)} f'(\bar{a}_n^\ell)},$$

for $n = 1, \dots, N$, because

$$\underline{\partial_n x_j^\ell(\bar{a}^\ell)} = \begin{cases} 0, & \text{if } j < n, \\ \Phi e^{-\alpha\theta(j-n)} f'(\bar{a}_n^\ell), & \text{if } j \geq n. \end{cases}$$

Inserting and into yields Inserting and into yields

(iv)-(vi) These statements follow immediately from the chain rule:

$n = 1, \dots, N$.

Step 4: Knowing the error signals, we can compute the weight gradient, i.e., the partial derivatives of the loss function w.r.t. the training parameters. For the partial derivatives of with respect to w.r.t. the output weights, we obtain

$$\frac{\partial \mathcal{E}}{\partial w_{pn}^{\text{out}}} \frac{\partial \mathcal{E}(\mathcal{W})}{\partial w_{pn}^{\text{out}}} = \frac{\partial_p \mathcal{E}(a^{\text{out}})}{\partial a_p^{\text{out}}} \frac{\partial \mathcal{E}}{\partial a_p^{\text{out}}} \frac{\partial a_p^{\text{out}}}{\partial w_{pn}^{\text{out}}} = \underline{\Delta} \delta_p^{\text{out}} x_n^L, \quad (47)$$

for $n = 1, \dots, N + 1$, $p = 1, \dots, P$.

For the partial derivatives with respect to w.r.t. the hidden weights, it holds

$$\frac{\partial \mathcal{E}}{\partial w_{nj}^\ell} \frac{\partial \mathcal{E}(\mathcal{W})}{\partial w_{nj}^\ell} = \frac{\partial_n \mathcal{E}(a^\ell)}{\partial a_n^\ell} \frac{\partial \mathcal{E}}{\partial a_n^\ell} \frac{\partial a_n^\ell}{\partial w_{nj}^\ell} = \underline{\Delta} \delta_n^\ell x_j^{\ell-1}, \quad (48)$$

for $j = 1, \dots, N + 1$, $n = 1, \dots, N$.

For the partial derivatives with respect to w.r.t. the input weights, the chain rule implies

$$\frac{\partial \mathcal{E}}{\partial w_{nm}^{\text{in}}} \frac{\partial \mathcal{E}(\mathcal{W})}{\partial w_{nm}^{\text{in}}} = \frac{\partial_n \mathcal{E}(a^1)}{\partial a_n^1} \frac{\partial \mathcal{E}}{\partial a_n^1} \frac{\partial a_n^1}{\partial w_{nm}^{\text{in}}} = \underline{\Delta} \delta_n^1 g'(a_n^{\text{in}}) \frac{\partial a_n^1}{\partial w_{nm}^{\text{in}}} \frac{\partial a_n^{\text{in}}}{\partial w_{nm}^{\text{in}}} = \underline{\Delta} \delta_n^1 g'(a_n^{\text{in}}) u_m, \quad (49)$$

for $m = 1, \dots, M + 1$, $n = 1, \dots, N$. This completes the proof.

The formulas for $\underline{\Delta}_n^\ell$, sums in Eq. (44) and Eq. (45), contain weighted sums over the hidden weights. These sums can be substituted as follows: (45) can be rewritten as sums over the index d of the delays:

$$\sum_{j=1}^N \sum_{i=1}^N e^{-\alpha \theta j} \delta_i^{\ell+1} w_{ij}^{\ell+1} = \sum_{\substack{d=1 \\ 1 \leq i-n'_d \leq N}} \sum_{\substack{d=1 \\ 1 \leq n+n'_d \leq N}}^D e^{-\alpha \theta (i-n'_d)} \delta_{n+n'_d}^{\ell+1} v_{d,i}^{\ell+1}. \quad (50)$$

and similarly

$$\sum_{j=n}^N e^{-\alpha \theta (j-n)} w_{ij}^{\ell+1} = \sum_{\substack{d=1 \\ n \leq i-n'_d \leq N}}^D e^{-\alpha \theta (i-n'_d-n)} v_{d,i}^{\ell+1}.$$

The equations follow from the definition of the hidden weight matrices. This way we achieve a substantially faster computation if the number of delays D is much smaller than the number of nodes per hidden layer N . Equation (50) is obtained by exploiting the special sparsity structure of the weight matrices W^ℓ , $\ell = 2, \dots, L$. The entries of these matrices are defined by Eq. (20), which yields we rewrite here using the indices of $w_{ij}^{\ell+1}$ from Eq. (50):

$$w_{i,i-n'_d}^{\ell+1} = \delta_{N+1,n} b_i^{\ell+1} + \sum_{d=1}^D \delta_{i-n'_d,n} v_{d,i}^{\ell+1} = \mathcal{M}_d^\ell(i\theta)_{d,n+n'_d}^{\ell+1}. \quad (51)$$

and $w_{ij}^\ell = 0$ for all $j \notin \{i - n'_d \mid d = 1, \dots, D\} \cup \{N + 1\}$. Replacing the sums over the index j by sums over the Since we have $1 \leq n \leq N$ in Eq. (44) and Eq. (45), the weight $w_{in}^{\ell+1}$ is non-zero only if there is an index $d \in 1, \dots, D$ such that $i - n'_d = n$, or equivalently $i = n + n'_d$. In this case we have $w_{in}^{\ell+1} = v_{d,i}^{\ell+1} = v_{d,n+n'_d}^{\ell+1}$. On the contrary, for any index d reduces the computational costs of the formulas significantly if $D \ll N$, the value $v_{d,n+n'_d}^{\ell+1}$ defines a matrix element of $W^{\ell+1}$ if and only if $1 \leq n + n'_d \leq N$. This implies Eq. (50).

4 Time signals of the Fit-DNN

Figure 8 illustrates how the Fit-DNN processes information by showing its time signals. Panel (a) illustrates the processing of process of obtaining the data signal $J(t)$ from an input image from the MNIST dataset, in this case an image of the handwritten number 4, to obtain the data signal $J(t)$, which is a step function with step size θ . First, the extended input vector u is multiplied by the trained input matrix W^{in} . Then an input preprocessing function g is applied element-wise to the entries of the obtained vector. The resulting values are the step heights of the data signal $J(t)$.

Panel (b) shows the internal processes in the hidden layers. From top to bottom we plot:

- the state of the system $x(t)$,
- the signal $a(t)$,
- the signal $a(t)$ decomposed into the positive and the negative parts of its components (i.e., the data signal, the modulated feedback signals, and the bias signal) indicated by their corresponding color,
- the data signal $J(t)$,
- the delayed feedback signals $x(t - \tau_d)$ (grey),
- the trained modulation functions $M_d(t)$ (colored),
- and the bias $b(t)$.

The signal $a(t)$ for the first hidden layer, $0 \leq t \leq T$, coincides with the data signal $J(t)$. For the remaining hidden layers, the signal $a(t)$ is a sum of the modulated feedback signals and the bias.

Panel (c) illustrates the output layer. The vector x_L , containing the values of $x(t)$ sampled at $t = (L-1)T + \theta, \dots, (L-1)T + N\theta$, is multiplied by the trained output matrix W^{out} to obtain the output activation vector. Then the softmax function is applied to obtain the output vector y^{out} . In this case, the Fit-DNN correctly identifies the input ~~image as~~ as an image showing the number 4.

The training process, which leads to the trained system depicted in Fig. 8, is shown in a video, which is attached as additional Supplementary Information.

References

- [1] Krizhevsky, A. Learning multiple layers of features from tiny images. *University of Toronto* (2012).
- [2] Netzer, Y. *et al.* Reading digits in natural images with unsupervised feature learning. *NIPS* (2011).
- [3] Lecun, Y., Bottou, L., Bengio, Y. & Haffner, P. Gradient-based learning applied to document recognition. *Proceedings of the IEEE* **86**, 2278–2324 (1998).
- [4] Xiao, H., Rasul, K. & Vollgraf, R. Fashion-mnist: a novel image dataset for benchmarking machine learning algorithms. Preprint at <https://arxiv.org/abs/1708.07747> (2017).
- [5] Hale, J. K. & Lunel, S. M. V. *Introduction to Functional Differential Equations* (Springer, New York, 1993).
- [6] Diekmann, O., Verduyn Lunel, S. M., van Gils, S. A. & Walther, H.-O. *Delay Equations* (Springer, New York, 1995).
- [7] Wu, J. *Introduction to Neural Dynamics and Signal Transmission Delay* (Walter de Gruyter, Berlin, Boston, 2001).
- [8] Erneux, T., Javaloyes, J., Wolfrum, M. & Yanchuk, S. Introduction to Focus Issue: Time-delay dynamics. *Chaos: An Interdisciplinary Journal of Nonlinear Science* **27**, 114201 (2017).
- [9] Hart, J. D., Schmadel, D. C., Murphy, T. E. & Roy, R. Experiments with arbitrary networks in time-multiplexed delay systems. *Chaos: An Interdisciplinary Journal of Nonlinear Science* **27**, 121103 (2017).
- [10] Hart, J. D., Larger, L., Murphy, T. E. & Roy, R. Delayed dynamical systems: networks, chimeras and reservoir computing. *Philosophical Transactions of the Royal Society A: Mathematical, Physical and Engineering Sciences* **377**, 20180123 (2019).
- [11] Bishop, C. M. *Pattern Recognition and Machine Learning* (Springer, New York, 2006).
- [12] Goodfellow, I., Bengio, Y. & Courville, A. *Deep Learning* (MIT Press, Cambridge, Massachusetts, London, England, 2016).

Figure 8: Time signals of the Fit-DNN after training. Panel (a) illustrates the processing of an input image to obtain the data signal $J(t)$. Panel (b) shows the internal processes of the hidden layers: the state variable of the system $x(t)$, and the signal $a(t)$, which consists of the data signal $J(t)$, the delayed feedback signals $x(t - \tau_d)$ multiplied by the modulation functions $M_d(t)$, and the bias signal $b(t)$. Panel (c) illustrates the output layer.

Reviewers' Comments:

Reviewer #1:

Remarks to the Author:

I thank the authors for carefully considering my comments in the revised version. In my opinion, this version of the paper discusses well its contributions and limitations. I believe that the core idea may have the potential of laying the ground for a new different line of research and I recommend acceptance.

Reviewer #2:

Remarks to the Author:

My comments have been satisfactorily addressed and I now recommend publication of the manuscript.

Andreas Amann

Reviewer #3:

Remarks to the Author:

I have read with great interest the answer of the authors to my queries, as well as the answers to the other reviewers. I am satisfied by this feedback and recommend publication as is.

Response to the referee reports:

Reviewer #1 (Remarks to the Author):

I thank the authors for carefully considering my comments in the revised version. In my opinion, this version of the paper discusses well its contributions and limitations. I believe that the core idea may have the potential of laying the ground for a new different line of research and I recommend acceptance.

Reviewer #2 (Remarks to the Author):

**My comments have been satisfactorily addressed and I now recommend publication of the manuscript.
Andreas Amann**

Reviewer #3 (Remarks to the Author):

I have read with great interest the answer of the authors to my queries, as well as the answers to the other reviewers. I am satisfied by this feedback and recommend publication as is.

We thank the reviewers for the positive evaluation of the manuscript and their recommendations for publication.